# MEGA: Message Passing Neural Networks for Multigraphs with EdGe Attributes

## Abstract

Edge-attributed multigraphs, in which multiple edges with distinct attributes connect the same pair of nodes, arise naturally in many real-world systems. In these graphs, effective learning requires preserving information from repeated interactions while distinguishing contributions from different neighbors. Existing neural network solutions for edge-attributed multigraphs remain limited: some lose information from repeated interactions, while others break permutation equivariance. To address this, we introduce *neighbor-aware aggregation*, an operator that first combines multi-edge features for each neighbor and then aggregates across neighbors. This operator captures per-neighbor statistics that standard single-stage aggregation cannot represent. Building on this operator, we present MEGA-GNN, a model-agnostic message-passing framework for edge-attributed multigraphs. We show that MEGA-GNN is permutation equivariant and has the same asymptotic complexity as standard GNNs with edge updates. We evaluate our approach on datasets from social networks and financial transaction networks. Neighbor-aware aggregation consistently improves GNN performance and matches or surpasses state-of-the-art methods.

## 1 Introduction

Graph Neural Networks (GNNs) have emerged as a powerful framework for learning on relational data, achieving strong performance across social networks, biological networks, recommendation systems, and knowledge bases (Hamilton et al., 2017; Xu et al., 2019; Gilmer et al., 2017; Corso et al., 2020; Veličković et al., 2018; Schlichtkrull et al., 2018; Vashishth et al., 2020; Chen et al., 2021b). Their success stems from the ability to learn node and edge representations that integrate the graph structure with rich attributes.

In many real-world systems, such as financial transaction, social interaction, and transportation networks, graphs are naturally represented as multigraphs. In such systems, the same pair of entities interact repeatedly, resulting in multiple edges, henceforth referred to as *multi-edges*, between the same pair of nodes, each with distinct attributes. Although standard GNNs can be readily applied to edge-attributed multigraphs, their theoretical expressiveness and practical performance remain limited in this setting (Egressy et al., 2024).

Accurately modeling multigraphs with rich edge attributes is critically important. Consider a financial transaction network in which a receiver obtains funds from multiple senders. Figure 1(a) illustrates such a setting: the target node receives messages from multiple neighbors, with some neighbors connected to it by multiple edges. Identifying which sender contributes the largest cumulative amount requires first aggregating transactions separately for each sender and then comparing the resulting totals. An approach that aggregates all incoming transactions at once cannot, in general, distinguish the largest per-sender total from the single largest transaction overall. This example highlights the importance of capturing *per-neighbor statistics*, namely, statistics computed separately for each neighbor, in learning on edge-attributed multigraphs.

Even though multi-relational graph neural networks are widely used in applications involving multigraphs, including knowledge graphs and social networks (Yu et al., 2023; Ye et al., 2022), these methods assume *labeled* multigraphs, where each edge is associated with a relation type. In practice, such relation types may be unavailable and non-trivial to infer from edge attributes. Furthermore, standard multi-relational GNN

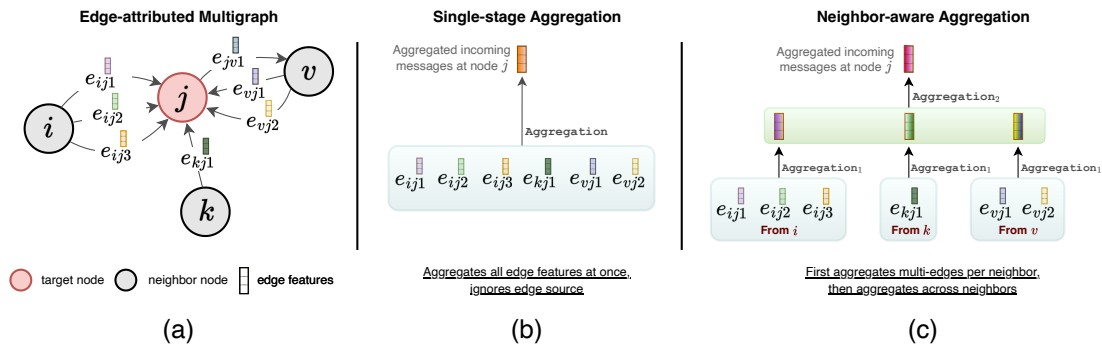

Figure 1: **Comparison of standard and neighbor-aware aggregation in edge-attributed multigraphs.** Standard GNNs aggregate all incoming edge features in a single step and therefore cannot distinguish between multi-edges and their sources. In contrast, neighbor-aware aggregation first aggregates multi-edges per neighbor and then aggregates across neighbors, explicitly preserving source information.

architectures (Schlichtkrull et al., 2018; Vashishth et al., 2020) lack mechanisms to distinguish between multi-edges and edges from different neighbors, thereby limiting their ability to compute per-neighbor statistics.

Recently, more specialized methods have also been proposed. Notably, Multi-GNN (Egressy et al., 2024) introduces adaptations for detecting arbitrary subgraph patterns in directed multigraphs. One key Multi-GNN adaptation is the introduction of auxiliary identifiers to distinguish edges connecting the same neighbor from those connecting different neighbors, which detrimentally breaks permutation equivariance. ADAMM (Sotiropoulos et al., 2023) preserves permutation equivariance by aggregating multi-edges into a single undirected edge representation before message passing. Similarly, DIAM (Ding et al., 2024) produces node embeddings by modeling the interactions of a node as a temporal sequence over its neighbors. However, both ADAMM and DIAM are designed to produce node-level representations; they do not maintain embeddings for individual edges and are therefore not suitable for edge-classification on edge-attributed multigraphs.

We introduce *neighbor-aware aggregation*, a novel operator for edge attributed multigraphs, illustrated in Figure 1(c), which separates aggregation over multi-edges from aggregation across neighbors. This is in contrast to the single-step aggregation mechanism of standard message passing GNNs, illustrated Figure 1(b). We demonstrate that neighbor-aware aggregation is strictly more expressive than the single-stage aggregation used in prior methods, as it captures per-neighbor statistics that single-stage aggregation cannot.

We further show that neighbor-aware aggregation is permutation invariant as long as the underlying aggregation functions are permutation invariant. In addition, neighbor-aware aggregation does not increase asymptotic complexity compared to GNNs with edge updates. This preservation of complexity is critical for scalability, as it ensures that the proposed aggregation operator can be applied to large-scale graphs without incurring additional computational bottleneck beyond those already present in existing GNN frameworks.

Building on neighbor-aware aggregation, we propose MEGA-GNN, a backbone-agnostic framework that extends standard GNNs to edge-attributed multigraphs. MEGA-GNN also supports bi-directional message passing for directed multigraph processing. We evaluate MEGA-GNN on user-state change prediction over social networks, anti-money laundering on financial transaction networks, and phishing account detection on the Ethereum blockchain. Across these benchmarks, neighbor-aware aggregation improves predictive performance over its corresponding GNN baselines and matches or surpasses state-of-the-art solutions.

## 2 Related Work

GNNs for multi-relational graphs and edge-attributed multigraphs form the closest line of work. In the multi-relational setting, methods such as R-GCN (Schlichtkrull et al., 2018), CompGCN (Vashishth et al., 2020), R-HGNN (Yu et al., 2023), and r-GAT (Chen et al., 2021b) incorporate relation types into message passing through mechanisms such as relation-dependent parameters, relation-specific subgraphs, or relation-aware attention mechanisms. From a theoretical perspective, Barcelo et al. (2022) characterize the expressive limits

of R-GCN and CompGCN and propose the more expressive higher-order k-RN architecture. However, these methods assume a fixed set of relation types and do not explicitly distinguish aggregation over multi-edges from aggregation across distinct neighbors, thereby limiting their ability to compute per-neighbor statistics.

Multi-GNN (Egressy et al., 2024) is a graph neural network specifically designed for edge-attributed multigraphs. It extends baseline GNNs with reverse message passing, port numbering, and ego IDs, yielding a provably more expressive architecture capable of detecting subgraph patterns in directed multigraphs. However, its port-numbering mechanism relies on precomputed edge identifiers and therefore does not preserve permutation equivariance (see Appendix A.2). Multi-FraudGT (Lin et al., 2024) incorporates these adaptations into a graph transformer architecture and therefore also does not preserve permutation equivariance.

ADAMM (Sotiropoulos et al., 2023) addresses graph-level anomaly detection in directed, edge-attributed multigraphs. It aggregates multi-edges into a single super-edge using a DeepSets-style encoder (Zaheer et al., 2017) as a preprocessing step, thereby preserving permutation equivariance. DIAM (Ding et al., 2024) models Ethereum transaction network as directed, edge-attributed multigraphs and learns node representations from temporal transaction patterns. However, both methods are designed to produce node-level representations for node- or graph-level tasks and do not maintain explicit embeddings for individual edges, making them unsuitable for edge-classification tasks. A detailed comparison of these methods is provided in Appendix A.2.

The expressive power of message passing architectures can be understood from several complementary perspectives, including graph-structural distinguishability via isomorphism tests (Xu et al., 2019; Morris et al., 2019; Maron et al., 2019; Barceló et al., 2021), as well as how information is aggregated within local neighborhoods and propagated across the graph (Fuchs* & Veličković*, 2023; Rosenbluth et al., 2023; Loukas, 2020). In edge-attributed multigraphs, neighborhoods exhibit additional structure, with multiple attributed interactions between the same node pairs. Standard single-stage aggregation ignores this structure. Our work shows that separating aggregation over multi-edges from aggregation across distinct neighbors yields a strictly more expressive architecture under moment-based aggregators, enabling richer local computations.

Aggregation design is another relevant line of work. Dehmamy et al. (2019); Corso et al. (2020) showed that combining multiple aggregators improves the empirical performance of GNNs. Kortvelesy et al. (2023) further proposed a learnable aggregator that can approximate standard multiset aggregators. Our neighbor-aware aggregation is compatible with these approaches and can incorporate them in a multi-level setting, first across edges for each neighbor pair and then across neighbors.

Multi-level aggregation has also been explored in other settings: Hypergraph GNNs (Feng et al., 2019; Huang & Yang, 2021; Gao et al., 2022) aggregate node features within each hyperedge and then across hyperedges, and P-GNN (You et al., 2019) aggregates first within anchor sets and then across anchor sets. Unlike our approach, these methods do not capture multiple attributed interactions between the same pair of nodes.

Graph machine learning has been increasingly applied to financial crime, including anti-money laundering in transaction networks (Weber et al., 2019; Cardoso et al., 2022; Altman et al., 2023), fraud and illicit account detection in payment or blockchain networks (Tam et al., 2019; Dou et al., 2020; Hiroki Kanezashi & Hirofuchi, 2022; Lin et al., 2024), and phishing or scam detection on Ethereum (Wu et al., 2020; Li et al., 2021b; Hu et al., 2023; Poursafaei et al., 2021). These studies demonstrate the strong potential of graph machine learning methods in the financial crime analysis domain, providing the motivation for our work.

## 3 Neighbor-Aware Message Passing on Edge-Attributed Multigraphs

This section introduces our notation as well as our neighbor-aware aggregation scheme. We first formally define the single-stage and our proposed neighbor-aware aggregation schemes. We then theoretically prove that the neighbor-aware aggregation is strictly more powerful than standard single-stage aggregation on multigraphs with edge attributes. We then present MEGA-GNN as a concrete instantiation of neighbor-aware aggregation operator within message passing layers. We further enhance MEGA-GNN with bi-directional message passing for directed multigraphs. Finally, we provide additional theoretical properties of our method, such as computational and memory complexity, and permutation equivariance.

### 3.1 Notation

**Definition 1** (Multiset). *A multiset is a 2-tuple $X = (S, m)$ where $S$ is the underlying set of $X$ formed from its distinct elements, and $m : S \to \mathbb{N}_{\geq 1}$ gives the multiplicity of the elements.*

**Definition 2** (Multiset Sum $\uplus$). *Let $A = (S_A, m_A)$ and $B = (S_B, m_B)$ be multisets over a common universe $U$. Their* sum *$A \uplus B$ is the multiset $C = (S_C, m_C)$ defined by*

$$S_C = S_A \cup S_B, \qquad m_C(x) = m_A(x) + m_B(x) \quad \text{for all } x \in U.$$

*Here, the operator $+$ denotes standard integer addition of multiplicities.*

**Definition 3** (Permutation Equivariance). *A function $\psi$ is permutation equivariant with respect to node and edge permutations if, for any permutation $\rho$ acting on the nodes and edges of a graph $\mathcal{G} = (\mathcal{V}, \mathcal{E})$, the following holds: $\psi(\rho \circ \mathcal{G}(\mathcal{V}, \mathcal{E})) = \rho \circ \psi(\mathcal{G}(\mathcal{V}, \mathcal{E}))$.*

We denote multisets with $\{\{\cdot\}\}$ and sets with $\{\cdot\}$. $[n]$ stands for the set $\{1, 2, \ldots, n\}$ for $n \in \mathbb{N}$. Let $\mathcal{G} = (\mathcal{V}, \mathcal{E})$ be a directed multigraph with node set $\mathcal{V}$ and edge multiset $\mathcal{E} = \{\{(i, j) \mid i, j \in \mathcal{V}\}\}$, where each $(i, j)$ represents a directed edge from node $i$ to node $j$. Let $\mathcal{E}^{\text{supp}} \subseteq \mathcal{E}$ denote the support set of $\mathcal{E}$. We define the edge multiplicity $P_{ij} := m_{\mathcal{E}}(i, j)$, i.e. the number of edges from node $i$ to node $j$. For a node $j \in \mathcal{V}$, the incoming and outgoing neighbors are defined as $N_{\text{in}}(j) = \{i \in \mathcal{V} \mid (i, j) \in \mathcal{E}^{\text{supp}}\}$ and $N_{\text{out}}(j) = \{i \in \mathcal{V} \mid (j, i) \in \mathcal{E}^{\text{supp}}\}$. We consider attributed multigraphs with feature dimensions $d_n, d_e, d \in \mathbb{N}$. Each node has an initial feature vector $\mathbf{x}_i^{(0)} \in \mathbb{R}^{d_n}$, and each $p$-th edge from $i$ to $j$ has a feature vector $\mathbf{e}_{ijp}^{(0)} \in \mathbb{R}^{d_e}$ and $p \in [P_{ij}]$. At the $l$-th layer where $l \in [L]$ and $L$ is the total number of layers, the latent node and edge features are denoted $\mathbf{x}_i^{(l)} \in \mathbb{R}^d$ and $\mathbf{e}_{ijp}^{(l)} \in \mathbb{R}^d$.

Let $\mathcal{M}_d$ denote the space of multisets over $\mathbb{R}^d$, and let $\mathcal{M}(\mathcal{M}_d)$ denote the space of multisets of such multisets. Let $j \in \mathcal{V}$ be a target node and, $X_{ij} = \{\{\mathbf{e}_{ijp} \mid p \in [P_{ij}]\}\} \in \mathcal{M}_d$, denote the multiset of edge feature vectors from node $i$ to node $j$. The neighborhood of $j$ is then given as $\mathcal{X}_j = \{\{X_{ij} \mid i \in N_{\text{in}}(j)\}\} \in \mathcal{M}(\mathcal{M}_d)$.

Let $g_1, \ldots, g_k : \mathcal{M}_d \to \mathbb{R}^d$ be a collection of coordinate-wise aggregators. Analogous to PNA (Corso et al., 2020), we define an aggregation function, $f_\theta$, that applies each aggregator to the input, concatenates the results, and processes the concatenated vector through an MLP:

$$f_\theta : \mathcal{M}_d \to \mathbb{R}^{d'}, \quad f_\theta(X) := \text{MLP}_\theta([g_1(X) \parallel \ldots \parallel g_k(X)]), \tag{1}$$

where $X \in \mathcal{M}_d$, $\parallel$ is concatenation, $\text{MLP}_\theta : \mathbb{R}^{kd} \to \mathbb{R}^{d'}$ is a feedforward network and $d'$ denotes the output dimension of the MLP.

### 3.2 Single-stage vs Neighbor-Aware Aggregation

In many real-world graphs, such as financial transaction networks, multiple edges may connect the same pair of nodes, each with distinct attributes. Standard GNNs, however, typically ignore this edge multiplicity and apply *single-stage aggregation*, which aggregates all incoming edges at once.

**Definition 4** (Single-stage Aggregation). *A single-stage aggregation function $\mathcal{T}_{\text{single-stage}} : \mathcal{M}(\mathcal{M}_d) \to \mathbb{R}^d$ aggregates all edge features in the neighborhood $\mathcal{X}_j$, treating it as a single multiset.*

$$\mathcal{T}_{\text{single-stage}}(\mathcal{X}_j) := f_\theta \Big( \biguplus_{X_{ij} \in \mathcal{X}_j} X_{ij} \Big). \tag{2}$$

In standard GNNs, $f_\theta$ is commonly implemented using a single aggregation function $g$, i.e., $k = 1$ in Equation 1, where $g$ is typically chosen as SUM, MEAN, or MAX.

Crucially, in multigraphs, single-stage aggregation fails to distinguish between edges from the same neighbor and those from different neighbors. To address this, we propose a *neighbor-aware aggregation* scheme: first, features of multi-edges between the same node pair are aggregated; second, the resulting messages from distinct neighbors are aggregated at the node level.

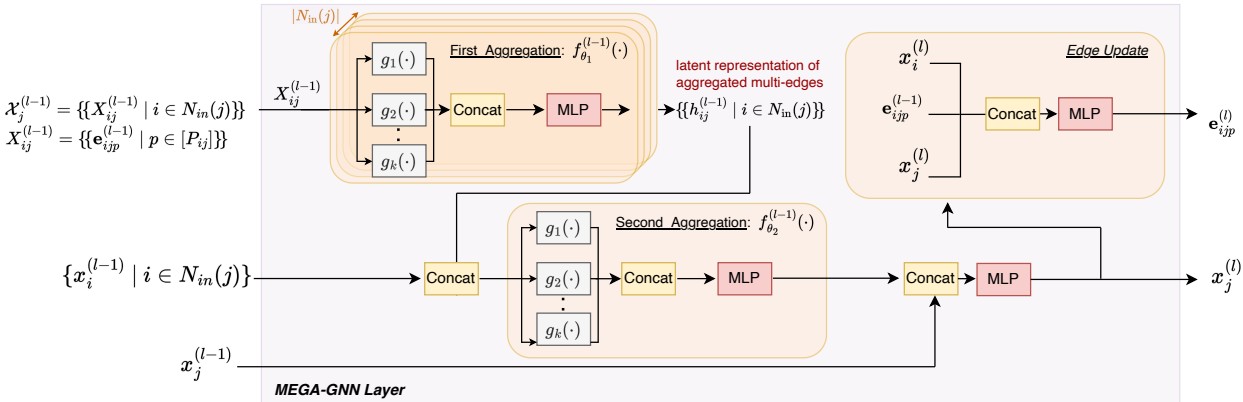

Figure 2: Overview of the MEGA-GNN layer. $g_1(\cdot), \ldots, g_k(\cdot)$ are aggregation functions.

**Definition 5** (Neighbor-aware Aggregation). *A neighbor-aware aggregation function $\mathcal{T}_{\text{neighb-aware}}$ : $\mathcal{M}(\mathcal{M}_d) \to \mathbb{R}^d$ first aggregates each $X_{ij} \in \mathcal{X}_j$ individually, and then aggregates the resulting multiset of vectors.*

$$\mathcal{T}_{\text{neighb-aware}}(\mathcal{X}_j) := f_{\theta_2}\left(\{\{f_{\theta_1}(X_{ij}) \mid X_{ij} \in \mathcal{X}_j\}\}\right), \tag{3}$$

This neighbor-aware aggregation operator naturally distinguishes edges based on their source nodes, enabling the computation of per-neighbor statistics. By applying multiple aggregators at both stages, as defined in Equation 1, we can extract more nuanced statistical information, capturing both per-neighbor and overall neighborhood characteristics. We formalize the enhanced representational capacity of this approach under a class of moment-based aggregators. Specifically, we employ the sum and raw moments to capture rich distributional features of multisets.

**Theorem 1.** *Neighbor-aware aggregation induces a strictly larger image than single-stage aggregation, if both schemes use the same set of $k$ aggregators: the sum and the raw moments of orders $2$ through $k$, defined as*

$$g_1(X) := \sum_{x \in X} x, \quad g_r(X) := \frac{1}{|X|} \sum_{x \in X} x^r, \quad 2 \le r \le k,$$

*where $X \in \mathcal{M}_d$ and $x^r$ is element-wise $r$-th power.*

The proof of Theorem 1 is in Appendix B.1. We first show that neighbor-aware aggregation can replicate single-stage aggregation by computing neighborhood moments from per-neighbor moments. Furthermore, we show that single-stage aggregation fails to capture per-neighbor moments, whereas neighbor-aware aggregation inherently does. For instance, statistics such as the variance across neighbors of their per-neighbor sums or the sum across neighbors of their per-neighbor variances are computable by neighbor-aware aggregation but not by single-stage aggregation.

### 3.3 The MEGA-GNN Architecture

This section presents MEGA-GNN as a concrete instantiation of *neighbor-aware aggregation* within a message-passing GNN framework for edge-attributed multigraphs.

Let $X_{ij}^{(l-1)} = \{\{\mathbf{e}_{ijp}^{(l-1)} \mid p \in [P_{ij}]\}\}$ denote the multiset of latent features of all edges from node $i$ to node $j$ at layer $(l-1)$. For each ordered node pair $(i, j)$ with at least one edge, MEGA-GNN first aggregates the features of these multi-edges:

$$\mathbf{h}_{ij}^{(l-1)} = f_{\theta_1}^{(l-1)}\left(X_{ij}^{(l-1)}\right), \tag{4}$$

where $f_{\theta_1}^{(l-1)} : \mathcal{M}_d \to \mathbb{R}^d$. The resulting vector $\mathbf{h}_{ij}^{(l-1)}$ summarizes the multi-edges from $i$ to $j$. These vectors are then aggregated over the incoming neighbors of node $j$:

$$\mathbf{a}_j^{(l-1)} = f_{\theta_2}^{(l-1)}\big(\{\{[\mathbf{x}_i^{(l-1)} \parallel \mathbf{h}_{ij}^{(l-1)}] \mid i \in N_{in}(j)\}\}\big), \tag{5}$$

$$\mathbf{x}_j^{(l)} = \phi_n^{(l-1)}\big([\mathbf{x}_j^{(l-1)} \parallel \mathbf{a}_j^{(l-1)}]\big), \tag{6}$$

where $\parallel$ denotes concatenation, $f_{\theta_2}^{(l-1)} : \mathcal{M}_{2d} \to \mathbb{R}^d$, and $\phi_n^{(l-1)} : \mathbb{R}^{2d} \to \mathbb{R}^d$ is the node update function at layer $(l-1)$. Finally, the latent features of each edge are updated:

$$\mathbf{e}_{ijp}^{(l)} = \phi_e^{(l-1)}\big([\mathbf{x}_i^{(l-1)} \parallel \mathbf{e}_{ijp}^{(l-1)} \parallel \mathbf{h}_{ij}^{(l-1)}]\big), \tag{7}$$

where $\phi_e^{(l-1)} : \mathbb{R}^{3d} \to \mathbb{R}^d$ is the edge update function.

Notably, MEGA-GNN preserves the original multigraph topology while enabling joint propagation of node and edge latent features at each layer. Unlike ADAMM (Sotiropoulos et al., 2023), it maintains distinct edges through individual updates. A detailed architecture diagram is provided in Figure 2.

### 3.4 MEGA-GNN with Bi-directional Message Passing

Bi-directional message passing improves model capacity by aggregating messages from incoming and outgoing neighbors separately. For example, it enables the computation of both the in-degree and the out-degree of a node, which is not possible using only incoming messages or by treating the graph as undirected (Egressy et al., 2024). This subsection describes the way MEGA-GNN implements bi-directional message passing in combination with two-stage aggregations.

Formally, we define reversed edges $(j, i)$ for each original edge $(i, j) \in \mathcal{E}$, and initialize their features as $\hat{\mathbf{e}}_{ijp}^{(0)} := \mathbf{e}_{jip}^{(0)}$, where $p \in [P_{ji}]$. At layer $(l-1)$, we denote the multiset of latent edge features from node $j$ to $i$ as $\hat{X}_{ij}^{(l-1)} = \{\{\hat{\mathbf{e}}_{ijp}^{(l-1)} \mid p \in [P_{ji}]\}\}$.

For each ordered node pair $(j, i)$ with at least one edge, MEGA-GNN first aggregates the features of the outgoing multi-edges from node $j$ to $i$.

$$\hat{\mathbf{h}}_{ij}^{(l-1)} = \hat{f}_{\theta_1}^{(l-1)}\big(\hat{X}_{ij}^{(l-1)}\big), \tag{8}$$

where $\hat{f}_{\theta_1}^{(l-1)} : \mathcal{M}_d \to \mathbb{R}^d$. The resulting vector $\hat{\mathbf{h}}_{ij}^{(l-1)}$ summarizes the multi-edges from $j$ to $i$. These vectors are then aggregated over the outgoing neighbors of node $j$:

$$\hat{\mathbf{a}}_j^{(l-1)} = \hat{f}_{\theta_2}^{(l-1)}\big(\{\{[\mathbf{x}_i^{(l-1)} \parallel \hat{\mathbf{h}}_{ij}^{(l-1)}] \mid i \in N_{out}(j)\}\}\big) \tag{9}$$

$$\mathbf{x}_j^{(l)} = \hat{\phi}_n^{(l-1)}\bigg([\mathbf{x}_j^{(l-1)} \parallel \mathbf{a}_j^{(l-1)} \parallel \hat{\mathbf{a}}_j^{(l-1)}]\bigg), \tag{10}$$

where $\hat{f}_{\theta_2}^{(l-1)} : \mathcal{M}_{2d} \to \mathbb{R}^d$, $\hat{\phi}_n^{(l-1)} : \mathbb{R}^{3d} \to \mathbb{R}^d$ is the node update function at layer $(l-1)$, and $\mathbf{a}_j^{(l-1)}$ is computed using Equation 5. Thus, messages from incoming and outgoing neighbors are aggregated separately and then combined to update the latent features of destination node $j$. Similarly, the latent features of the reverse edges are updated with function $\hat{\phi}_e^{(l-1)} : \mathbb{R}^{3d} \to \mathbb{R}^d$:

$$\hat{\mathbf{e}}_{ijp}^{(l)} = \hat{\phi}_e^{(l-1)}\big([\mathbf{x}_i^{(l-1)} \parallel \hat{\mathbf{e}}_{ijp}^{(l-1)} \parallel \hat{\mathbf{h}}_{ij}^{(l-1)}]\big), \tag{11}$$

A detailed architecture diagram of the proposed method with bi-directional message passing capability is provided in Appendix C.1.

### 3.5 Additional Properties of MEGA-GNN

We analyze the computational and structural properties of MEGA-GNN. In particular, we show that (i) the proposed architecture preserves the asymptotic computational and memory complexity of standard message-passing GNNs with edge updates, while (ii) maintaining permutation equivariance over multigraphs.

### 3.5.1 Computational and Memory Complexity

We analyze the computational and memory complexity of MEGA-GNN and show that the proposed neighbor-aware aggregation mechanisms do not introduce asymptotic overhead compared to standard message-passing GNNs.

**Theorem 2.** *The asymptotic per-layer forward-pass complexity of MEGA-GNN is*

$$O\big((|\mathcal{E}| + |\mathcal{V}|)d^2 + (|\mathcal{E}| + |\mathcal{V}|)d\big).$$

*Proof.* Using the notation in Section 3.1, let $d$ denote the node/edge embedding dimension, and assume all linear maps are $\mathbb{R}^d \to \mathbb{R}^d$ (incurring $\mathcal{O}(d^2)$ complexity). The first two groups correspond to aggregation over multi-edges and neighbors, respectively, each consisting of aggregation, linear transformation, and pointwise nonlinearity. The final group corresponds to the edge update, which involves a linear transformation followed by a nonlinearity. The per-layer complexity is

$$\mathcal{O}\big(\underbrace{|\mathcal{E}|\,d + |\mathcal{E}^{\text{supp}}|\,d^2 + |\mathcal{E}^{\text{supp}}|\,d}_{\text{aggregation over multi-edges}} + \underbrace{|\mathcal{E}^{\text{supp}}|\,d + |\mathcal{V}|\,d^2 + |\mathcal{V}|\,d}_{\text{aggregation over neighbors}} + \underbrace{|\mathcal{E}|\,d^2 + |\mathcal{E}|d}_{\text{edge update with non-linearity}}\big).$$

Since $|\mathcal{E}^{\text{supp}}| \leq |\mathcal{E}|$, the per-layer complexity simplifies to

$$\mathcal{O}\big((|\mathcal{E}| + |\mathcal{V}|)\,d^2 + (|\mathcal{E}| + |\mathcal{V}|)\,d\big).$$

$\square$

**Corollary 1.** *A single layer of MEGA-GNN achieves the same asymptotic computational complexity as a standard message-passing GNN with edge updates.*

*Proof.* A standard message-passing GNN layer with edge updates incurs

$$\mathcal{O}\big(\underbrace{|\mathcal{E}|\,d}_{\text{neighborhood agg.}} + \underbrace{|\mathcal{V}|\,d^2}_{\text{node update}} + \underbrace{|\mathcal{V}|\,d}_{\text{nonlinearity}} + \underbrace{|\mathcal{E}|\,d^2}_{\text{edge update}} + \underbrace{|\mathcal{E}|\,d}_{\text{nonlinearity}}\big).$$

Collecting terms yields

$$\mathcal{O}\big((|\mathcal{E}| + |\mathcal{V}|)\,d^2 + (|\mathcal{E}| + |\mathcal{V}|)\,d\big),$$

which matches the complexity of MEGA-GNN from Theorem 2. $\square$

**Theorem 3.** *For a single MEGA-GNN layer, the forward-pass memory complexity is*

$$\mathcal{O}(|\mathcal{V}|d + |\mathcal{E}|d).$$

Proof of Theorem 3 is given in Appendix B.2. Together, these results show that MEGA-GNN matches the computational and memory complexity of standard message-passing GNNs with edge updates, despite explicitly modeling multi-edges.

### 3.5.2 Permutation Equivariance

**Proposition 1** (Permutation Equivariance). *Given aggregation functions $f_{\theta_1}$ and $f_{\theta_2}$ that are permutation invariant over multisets, MEGA-GNN is permutation equivariant with respect to arbitrary permutations of nodes and edges in the input multigraph, including permutations over multi-edges.*

Proposition 1 (proof in Appendix B.3) establishes that MEGA-GNN preserves permutation equivariance at both node and edge levels. Notably, this property holds even in the presence of multi-edges, in contrast to Multi-GNN (Egressy et al., 2024), as discussed in Appendix A.2.

# 4 Experiments

We first present a controlled diagnostic experiment on a synthetic benchmark designed to test the ability of models to compute per-neighbor statistics in directed edge-attributed multigraphs. This experiment is intended to probe the core aggregation mechanism underlying MEGA-GNN. We then evaluate MEGA-GNN on ten datasets spanning two domains (social networks and financial transaction networks) and three prediction tasks. Each experiment is repeated at least five times with different random seeds; we report means and standard deviations throughout. In all experiments, we implement the models in PyTorch (2.2.2) and PyTorch Geometric (2.5.2) (Paszke et al., 2019; Fey & Lenssen, 2019). Additional implementation details are provided in Appendix C.

## 4.1 Controlled Diagnostic of Per-Neighbor Aggregation

**Datasets and Setup.** We evaluate on a synthetic node regression benchmark designed to isolate the ability of models to compute per-neighbor statistics in directed edge-attributed multigraphs. Each split is generated independently from a Barabási-Albert graph, whose edges are randomly oriented and then expanded into a multigraph by sampling a random number of multi-edges per node pair. Edge attributes are scalar amounts drawn from a log-normal distribution, while node features are constant. The task is to predict per-neighborhood statistics that depend on grouping incoming edges by their source nodes; labels are defined only for nodes with at least two distinct incoming sources. Targets include the maximum source total, variance of source totals, gap between the two largest source totals, sum of source-wise maxima, and standard deviation of source-wise maxima. Train, validation, and test splits are generated with different random seeds, yielding disjoint graphs with the same generation process. We compare PNA (Corso et al., 2020) and FraudGT (Lin et al., 2024) as strong message-passing and graph-transformer baselines, together with their Multi-GNN variants (Egressy et al., 2024). We further include ADAMM (Sotiropoulos et al., 2023). All baselines use the same hidden dimension size.

**Results.** Table 1 shows that MEGA-PNA achieves the lowest MAE, reducing the error of the strongest baseline (ADAMM) by 35.1%. Multi-GNN adaptations (Multi-PNA, Multi-FraudGT) improve over their base models, indicating the benefit of explicitly modeling multigraph structure, but remain well behind MEGA-PNA. ADAMM performs better than the other baselines, likely due to its DeepSets-style aggregation (Zaheer et al., 2017) of multi-edges. However, this is applied once as preprocessing, collapsing multi-edges into a single super-edge, rather than being integrated into each message-passing layer. These results are consistent with Theorem 1: baselines that aggregate over the full neighborhood in a single step cannot distinguish edges by their source nodes, and therefore fail to compute source-grouped statistics. MEGA-PNA's neighbor-aware aggregation retains this grouping, which is reflected in its lower error. Whether this capability translates into gains on real-world data is examined in Sections 4.2–4.3.

Table 1: Synthetic node-regression results on the per-neighbor statistics benchmark.

| Model | MAE ($\downarrow$) |
|---|---|
| PNA | $0.2556 \pm 0.0027$ |
| FraudGT | $0.3920 \pm 0.0078$ |
| ADAMM | $\underline{0.1559} \pm \underline{0.0038}$ |
| Multi-FraudGT | $0.3383 \pm 0.0097$ |
| Multi-PNA | $0.1975 \pm 0.0059$ |
| MEGA-PNA (ours) | $\mathbf{0.1012} \pm \mathbf{0.0066}$ |

## 4.2 Social Networks

**Datasets and Setup.** We evaluate user state-change prediction on three temporal user-item interaction datasets from JODIE (Kumar et al., 2019): Reddit bans, Wikipedia bans, and MOOC student dropouts. The goal is to predict whether an interaction is associated with a future state change in the user (ban or dropout). Interactions are labeled 0 until the event occurs, and the final interaction before the event is labeled 1. This is a highly imbalanced setting as shown in Table 11 in the Appendix C.4. Following Kumar et al. (2019), we use a 60/20/20 temporal split. We compare our MEGA variants with GIN (Xu et al., 2019) and PNA (Corso et al., 2020) against JODIE (Kumar et al., 2019), the base GNNs, and their Multi-GNN (Egressy et al., 2024) variants. We adopt ego IDs (You et al., 2021) and bidirectional message passing, which are also used in Multi-GNNs.

Table 2: User state-change prediction performance (ROC-AUC, %) on three temporal user-item interaction datasets from social and collaborative platforms.

| Method | MOOC | Wikipedia | Reddit |
|---|---|---|---|
| JODIE (Kumar et al., 2019) | 75.6 ± na | 83.1 ± na | 59.9 ± na |
| GIN | 66.2 ± 1.2 | 82.0 ± 1.3 | 56.9 ± 1.3 |
| Multi-GIN | 73.7 ± 2.3 | 90.3 ± 0.5 | 65.5 ± 1.6 |
| MEGA-GIN (ours) | 75.3 ± 3.7 | 89.8 ± 1.2 | **72.7** ± 1.5 |
| PNA | 66.3 ± 3.3 | 76.7 ± 2.5 | 63.5 ± 5.9 |
| Multi-PNA | 70.1 ± 3.6 | 90.4 ± 1.3 | 61.6 ± 5.7 |
| MEGA-PNA (ours) | **76.1** ± 1.8 | **92.1** ± 0.9 | 67.6 ± 2.8 |

**Results.** Table 2 shows that MEGA-GNN consistently improves both GIN and PNA across the three JODIE datasets, with the best MEGA variant attaining the highest ROC-AUC on every dataset. Multi-GNN already improves over the base GNNs, confirming that explicitly modeling multigraph structure is beneficial; neighbor-aware aggregation then yields further gains over Multi-GNN. The largest improvements occur on Reddit, the most imbalanced dataset, suggesting that neighbor-aware aggregation is particularly beneficial in highly imbalanced settings.

### 4.3 Financial Transaction Networks

We tackle money laundering detection via edge classification on six synthetic transaction datasets (Altman et al., 2023) and phishing account detection via node classification on a real-world ethereum blockchain graph (Chen et al., 2021a). Table 12 in the Appendix C.4 summarizes the dataset statistics; dataset-specific train/validation/test split strategies are also detailed in Appendix C.4.

#### 4.3.1 AML Edge Classification

**Datasets and Setup.** We evaluate money laundering detection on IBM's realistic synthetic Anti-Money Laundering (AML) benchmark (Altman et al., 2023), comprising six datasets: small, medium, and large variants, each in a low-illicit (LI, ∼0.05%) and high-illicit (HI, ∼0.1%) version. The small, medium, and large datasets contain approximately 6M, 30M, and 180M transactions, respectively. Datasets are modelled as directed edge-attributed multigraphs where nodes represent accounts and edges represent transactions. Edges carry four attributes: timestamp, amount, currency, and payment format, and the task is edge classification, labeling each transaction as licit or illicit.

We evaluate four GNN baselines, GIN (Xu et al., 2019), PNA (Corso et al., 2020), GenAgg (Kortvelesy et al., 2023), and R-GCN (Schlichtkrull et al., 2018), and a graph transformer baseline FraudGT (Lin et al., 2024) in combination with two multigraph adaptations: Multi-GNN (Egressy et al., 2024) and MEGA-GNN. Base models do not use multigraph adaptations but leverage edge updates by default. We use the same temporal splits as Multi-GNN. We adopt ego IDs (You et al., 2021), and incorporate bi-directional message passing, which are also used in Multi-GNN.

We extend R-GCN to support edge attributes and edge updates, terming this variant R-GCNE (Appendix C.6). On AML, edge types are derived from transaction currency, so edges carry both relation types and attributes. When both forms of information are available, MEGA-GNN can be combined with R-GCNE to form MEGA-R-GCNE. In these experiments, R-GCNE is extended to support only MEGA adaptations. MEGA can be paired with any baseline (see Appendix C.5).

**Results.** Figure 3 reports minority-class F1 scores across the six AML datasets, while Table 15 in Appendix E.1 provides the complete numerical results in tabular form.

Focusing on the comparisons shown in Figure 3, MEGA-GNNs outperform their corresponding baselines in all 24 cases. MEGA-GNNs also outperform Multi-GNNs in 17 of 18 head-to-head comparisons, with average minority-class F1 gains of 4.62 on HI datasets and 6.77 on LI datasets. These consistent gains across backbones suggest that improvements come from MEGA-GNN's neighbor-aware aggregation rather than

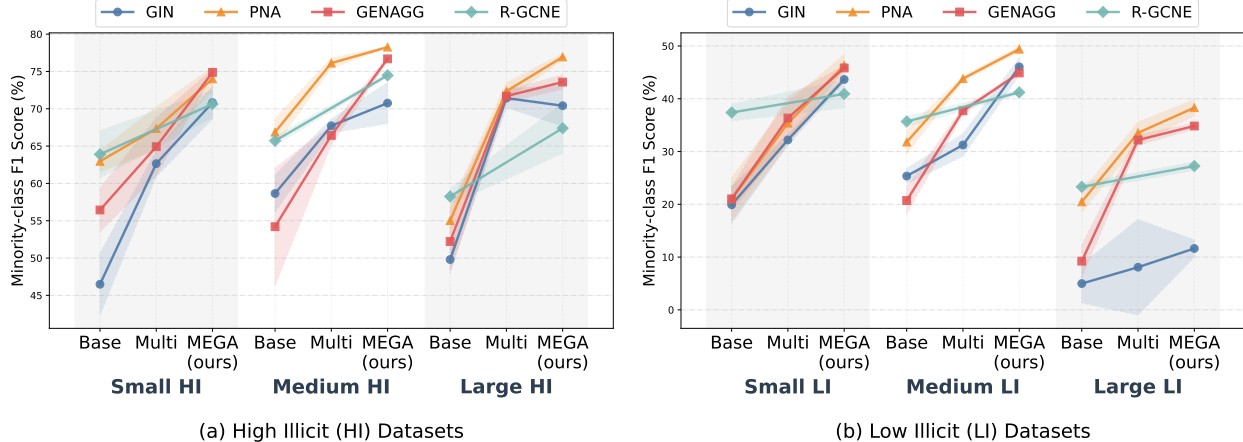

Figure 3: Minority class F1 scores (%) for six AML datasets using four different GNN baselines (GIN, PNA, GenAgg, and R-GCNE) and two different multigraph adaptations (Multi and MEGA). This view isolates the effect of the multigraph adaptation; Table 15 in the Appendix E.1 lists full F1 scores.

Table 3: ETH node classification: minority-class F1 scores (%). The best result is given in **boldface**.

| | ADAMM | | Multi | | | DIAM | MEGA (ours) | |
| | GIN | PNA | GIN | PNA | FraudGT | | GIN | PNA |
|---|---|---|---|---|---|---|---|---|
| **F1** | $34.73 \pm 15.75$ | $37.99 \pm 5.41$ | $51.34 \pm 3.92$ | $\underline{64.61 \pm 1.40}$ | $57.40 \pm 0.91$ | $64.43 \pm 1.07$ | $57.45 \pm 1.14$ | $\mathbf{64.84 \pm 1.73}$ |

architecture-specific effects. As formalized in Theorem 1, MEGA-GNN can compute per-neighbor statistics that standard GNNs cannot. The larger gains on LI datasets are consistent with this mechanism: illicit activity in LI datasets is more concentrated on node pairs connected by multiple transactions than in the corresponding HI datasets (Table 13, Appendix C.4). We further analyze this pattern in Appendix E.5.

MEGA adaptations also consistently improve R-GCNE, indicating that the two approaches are orthogonal and complementary: R-GCNE handles labeled multigraphs with predefined relation types, while MEGA-GNN targets edge-attributed multigraphs with high-dimensional features. When both forms of information are present, MEGA-R-GCNE (Figure 6 in Appendix A.1) effectively leverages them.

Table 4 and Table 15 (Appendix E.1) additionally compare MEGA-PNA with Multi-FraudGT. Multi-FraudGT is competitive, particularly on AML Small datasets; however, MEGA-PNA achieves the highest F1 scores on five of six datasets. Table 15 (Appendix E.1) further reports each model's average rank across the six datasets: MEGA variants occupy four of the top six positions, with MEGA-PNA achieving the best average rank (1.33), followed by MEGA-GenAgg (2.33) and Multi-FraudGT (2.83). Comparing MEGA-PNA directly against the strongest non-MEGA baseline on each dataset yields a mean improvement of +2.07 F1 points, with the single loss occurring on AML-Small-HI, where Multi-FraudGT is narrowly better. We additionally verify the statistical significance of these improvements via two-sided Wilcoxon signed-rank tests; full results are reported in Appendix E.4.

### 4.3.2 ETH Node Classification

**Datasets and Setup.** We evaluate phishing account detection on the Ethereum Phishing Transaction Network (ETH) (Chen et al., 2021a), a real-world ethereum blockchain graph with 2.97M accounts and 13.6M transactions. The network is modeled as directed edge-attributed multigraph where nodes represent accounts and edges represent transactions. Nodes are labeled as phishing or benign, edges carry timestamp and amount, and the task is node classification.

We evaluate GIN and PNA in combination with three multigraph adaptations, Multi-GNN (Egressy et al., 2024), ADAMM (Sotiropoulos et al., 2023), and MEGA-GNN. We additionally compare against Multi-

Table 4: Impact of permuting port numbers on the F1 scores (%) of Multi-PNA and Multi-FraudGT. "Rank" denotes the average rank of each model across all datasets, where rank 1 corresponds to the best performing model on a given dataset (lower is better). We use "(perm.)" to indicate that port numbers are permuted.

| Ablation | AML Small HI | AML Small LI | AML Medium HI | AML Medium LI | AML Large HI | AML Large LI | ETH | Rank (↓) |
|---|---|---|---|---|---|---|---|---|
| Multi-PNA | $67.35 \pm 2.89$ | $35.40 \pm 3.93$ | $\underline{76.13 \pm 0.69}$ | $43.82 \pm 0.51$ | $72.35 \pm 1.14$ | $33.54 \pm 2.04$ | $64.61 \pm 1.40$ | 2.71 |
| Multi-PNA (perm.) | $63.77 \pm 2.47$ | $31.48 \pm 0.72$ | $73.36 \pm 0.83$ | $43.24 \pm 0.24$ | $70.93 \pm 0.69$ | $32.18 \pm 1.72$ | $62.71 \pm 2.73$ | 3.85 |
| Multi-FraudGT | $\mathbf{75.81 \pm 0.75}$ | $\underline{45.69 \pm 1.14}$ | $75.97 \pm 0.18$ | $\underline{44.66 \pm 0.58}$ | $\underline{73.04 \pm 0.59}$ | $\underline{35.49 \pm 0.52}$ | $57.40 \pm 0.91$ | 2.28 |
| Multi-FraudGT (perm.) | $61.74 \pm 1.68$ | $30.15 \pm 2.67$ | $65.89 \pm 5.61$ | $32.05 \pm 1.35$ | $63.33 \pm 1.35$ | $29.95 \pm 1.18$ | $49.59 \pm 1.83$ | 5 |
| MEGA-PNA | $\underline{74.01 \pm 1.55}$ | $\mathbf{46.32 \pm 2.07}$ | $\mathbf{78.26 \pm 0.11}$ | $\mathbf{49.40 \pm 0.54}$ | $\mathbf{76.95 \pm 0.44}$ | $\mathbf{38.31 \pm 1.53}$ | $\mathbf{64.84 \pm 1.73}$ | 1.14 |

Table 5: Effect of artificially collapsing multi-edges on the ETH dataset. At each collapse rate, a fraction of multi-edges is merged into a single edge (mean amount, earliest timestamp). MEGA-PNA uses neighbor-aware aggregation only (no bi-directional MP) to isolate the effect of multi-edge structure. F1 scores are minority-class F1 (%); gap is MEGA-PNA minus PNA (pp).

| Collapse rate | 0% | 25% | 50% | 75% | 100% |
|---|---|---|---|---|---|
| PNA | $53.93 \pm 2.45$ | $53.95 \pm 3.48$ | $52.94 \pm 3.71$ | $53.92 \pm 4.68$ | $53.11 \pm 1.78$ |
| MEGA-PNA (PNA with neighbor-aware Agg.) | $59.13 \pm 0.51$ | $56.39 \pm 1.29$ | $56.37 \pm 2.42$ | $53.50 \pm 0.90$ | $52.67 \pm 1.66$ |
| Gap (pp) | $+5.20$ | $+2.44$ | $+3.43$ | $-0.42$ | $-0.44$ |

FraudGT (Lin et al., 2024), a graph transformer with Multi-GNN adaptations, and DIAM (Ding et al., 2024), a specialized ETH solution. We use a 65/15/20 temporal split and incorporate bi-directional message passing, which is also used in Multi-GNN

**Results.** Table 3 reports results on the ETH benchmark. MEGA-GIN improves minority-class F1 by 6.11% over Multi-GIN, and MEGA-PNA achieves the highest F1, slightly surpassing both Multi-PNA and DIAM. Compared to ADAMM, MEGA-GNN variants deliver over 20% higher F1. MEGA-PNA also outperforms Multi-FraudGT by 7.44%, confirming the effectiveness of neighbor-aware aggregation and bi-directional message passing on node-level tasks.

## 4.4 Further Comparisons and Ablations

In this section, we first study the robustness of Multi-PNA and Multi-FraudGT to permutations of their pre-computed port numbers, illustrating the effect of lacking permutation equivariance. We then probe whether MEGA-GNN's gains on real data are tied to multi-edge structure by artificially collapsing multi-edges on ETH. We next analyze the contribution of individual MEGA-GNN components including neighbor-aware aggregation, bi-directional message passing, and ego IDs, quantifying the standalone impact of neighbor-aware aggregation. Finally, we compare against capacity-matched baselines to disentangle inductive bias from parameter count.

**Robustness to port permutations.** Permutation equivariance of MEGA-GNN is established theoretically in Proposition 1. In contrast, Multi-PNA and Multi-FraudGT rely on precomputed port numbers and are therefore not permutation equivariant (Proposition 2, Appendix A.2); here we examine how this affects performance in practice. Concretely, we permute port numbers during test set evaluation. As shown in Table 4, MEGA-PNA outperforms both baselines in most settings. Permuting port numbers during test set evaluation substantially degrades Multi-PNA and Multi-FraudGT, further widening the performance gap.

**Multi-edge structure perturbation on real data.** To test whether MEGA-GNN's advantage on real data is specifically tied to multi-edge structure, we artificially collapse multi-edges on the ETH dataset at rates of 0%, 25%, 50%, 75%, and 100%. At each rate, a random subset of multi-edges between the same node pairs is replaced by a single edge whose amount is the group mean and whose timestamp is the group's earliest; all other graph properties are unchanged. We compare MEGA-PNA with neighbor-aware aggregation only (without bi-directional message passing) against the base PNA, so that multi-edge presence is the sole variable. Table 5 reports the results. PNA's performance is essentially flat across all collapse rates

Table 6: Impact of neighbor-aware aggregation, bi-directional message passing (MP) and ego IDs: minority-class F1 scores (%) of MEGA-GNN. Parameter counts are reported separately for AML (edge classification) and ETH (node classification) variants. K denotes $10^3$.

| Ablation | # params (AML) | AML Small HI | AML Small LI | AML Medium HI | AML Medium LI | # params (ETH) | ETH |
|---|---|---|---|---|---|---|---|
| GIN | 69.6K | $46.50 \pm 4.11$ | $19.93 \pm 3.55$ | $58.65 \pm 2.50$ | $25.36 \pm 1.49$ | 17.8K | $42.33 \pm 3.70$ |
| MEGA-GIN (GIN with neighbor-aware Agg.) | 86.3K | $69.98 \pm 2.02$ | $41.45 \pm 2.13$ | $69.50 \pm 0.85$ | $44.69 \pm 0.13$ | 22.1K | $43.56 \pm 2.67$ |
| MEGA-GIN w/ bi-directional MP | 161.2K | $\mathbf{72.50 \pm 3.26}$ | $\underline{41.67 \pm 1.51}$ | $\mathbf{74.40 \pm 0.87}$ | $\underline{44.81 \pm 0.48}$ | 41.1K | $\mathbf{57.45 \pm 1.14}$ |
| MEGA-GIN w/ ego IDs & bi-directional MP | 161.3K | $\underline{70.83 \pm 2.18}$ | $\mathbf{43.66 \pm 0.54}$ | $\underline{70.77 \pm 2.76}$ | $\mathbf{46.05 \pm 1.64}$ | 41.1K | $\underline{55.19 \pm 2.33}$ |
| PNA | 32.2K | $62.96 \pm 1.43$ | $21.02 \pm 4.05$ | $66.87 \pm 1.87$ | $31.79 \pm 2.30$ | 30.1K | $53.93 \pm 2.45$ |
| MEGA-PNA (PNA with neighbor-aware Agg.) | 41.8K | $73.65 \pm 0.36$ | $43.77 \pm 1.53$ | $76.77 \pm 0.19$ | $\underline{48.08 \pm 0.32}$ | 39.7K | $59.13 \pm 0.51$ |
| MEGA-PNA w/ bi-directional MP | 79.1K | $\mathbf{74.98 \pm 1.59}$ | $\underline{45.36 \pm 1.18}$ | $\underline{77.47 \pm 0.41}$ | $47.36 \pm 0.89$ | 77.1K | $\mathbf{64.84 \pm 1.73}$ |
| MEGA-PNA w/ ego IDs & bi-directional MP | 79.2K | $\underline{74.01 \pm 1.55}$ | $\mathbf{46.32 \pm 2.07}$ | $\mathbf{78.26 \pm 0.11}$ | $\mathbf{49.40 \pm 0.54}$ | 77.1K | $\underline{60.02 \pm 5.10}$ |

Table 7: Impact of neighbor-aware aggregation, bi-directional message passing, and ego IDs on inference throughput of MEGA-GNN. Throughput is reported as transactions/s for AML and accounts/s for ETH.

| Ablation | AML Small HI | AML Small LI | AML Medium HI | AML Medium LI | ETH |
|---|---|---|---|---|---|
| PNA | **277 829** | **254 654** | **135 218** | **137 103** | **1 125 842** |
| MEGA-PNA (PNA with neighbor-aware Agg.) | $\underline{236\,813}$ | $\underline{217\,623}$ | $\underline{118\,810}$ | $\underline{120\,022}$ | $\underline{852\,907}$ |
| MEGA-PNA w/ bi-directional MP | 56 251 | 51 278 | 27 293 | 27 167 | 362 749 |
| MEGA-PNA w/ ego IDs & bi-directional MP | 53 958 | 52 660 | 27 548 | 27 170 | 365 341 |

($\sim 53\%$ F1), consistent with single-stage aggregation being unable to exploit multi-edge structure even when it is present. In contrast, MEGA-PNA degrades from 59.13% at 0% collapse to 52.67% at full collapse, and its advantage over PNA shrinks from +5.20 pp to $-0.44$ pp. The disappearance of the gap precisely when multi-edge structure is removed indicates that MEGA-PNA's gains stem from the multi-edge structure it is designed to exploit, complementing the mechanism-level evidence in Section 4.1.

**Component ablations.** Table 6 shows the effect of each MEGA-GNN component for two different GNN baselines: GIN and PNA. In both cases, neighbor-aware aggregation yields the largest improvement over the base model, which is consistent with the theoretical result in Theorem 1. Bi-directional message passing brings further gains, especially on ETH. Ego IDs (You et al., 2021) are helpful in some cases, particularly on several AML low-illicit datasets, but their effect is not consistent across all settings.

**Capacity-matched comparison.** A natural concern is whether the gains from neighbor-aware aggregation reflect its inductive bias or simply the additional parameters it introduces. To disentangle these factors, we compare against capacity-matched GIN and PNA baselines, scaled to approximately match the parameter counts of MEGA-GIN and MEGA-PNA with neighbor-aware aggregation, respectively. Increasing baseline capacity yields only modest and inconsistent improvements, and in several cases slightly degrades performance, leaving the gap to MEGA variants largely intact. The results are in Table 18(Appendix E.2).

### 4.5 Inference Throughput and Scalability Analysis

In this subsection, we evaluate the computational performance and scalability of MEGA-PNA through two analyses: (i) inference throughput under architectural ablations, and (ii) scalability with respect to neighborhood size, measured in terms of inference throughput and peak GPU memory consumption. Throughput is measured as the number of processed instances per second (transactions for AML datasets and accounts for the ETH dataset) and includes both subgraph sampling time and the forward pass. All experiments were conducted on a single machine equipped with an NVIDIA GeForce RTX 4090 GPU (24 GB memory) and an AMD Ryzen 9 7950X 16-Core Processor, with 64 GB of system RAM. All models were implemented in PyTorch 2.2.2 with PyTorch Geometric 2.5.2 and executed using CUDA 11.8.

Table 7 details the inference throughput for the ablated MEGA-PNA variants. Integrating neighbor-aware aggregation into the base PNA architecture introduces only a modest reduction in throughput across all datasets. This overhead is consistent with our theoretical analysis in Section 3.5.1, which establishes that neighbor-aware aggregation preserves the asymptotic complexity of standard message-passing GNNs with edge updates. Considering the results in Table 7 together with the minority-class F1 scores in Table 6,

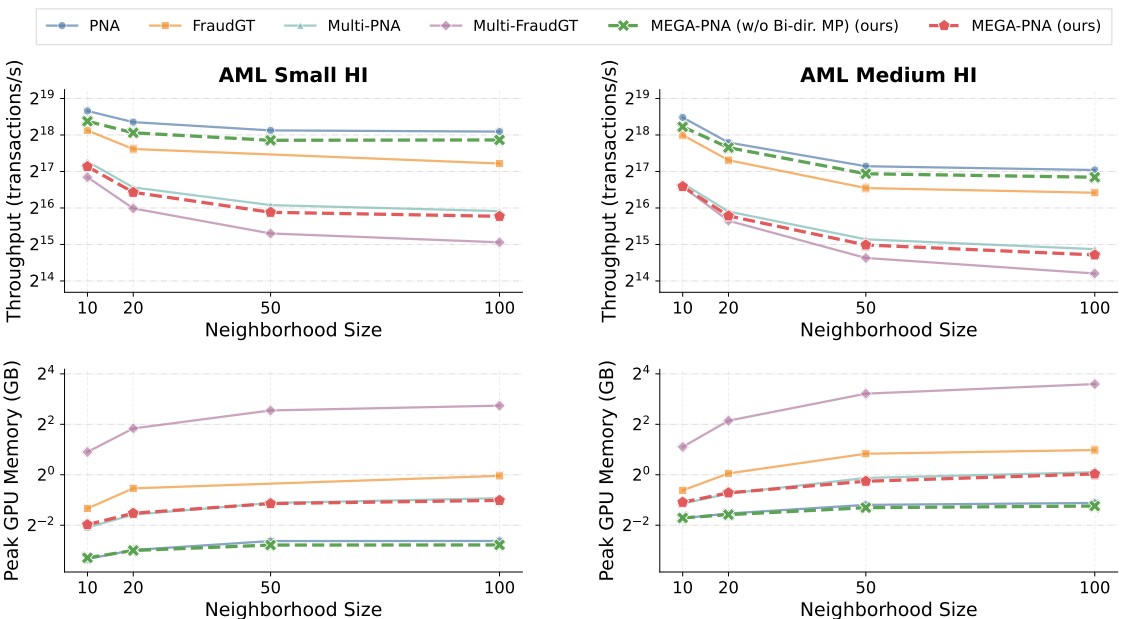

Figure 4: Throughput and memory usage vs. neighborhood size on two AML datasets of different scales.

we observe a clear computational trade-off. Neighbor-aware aggregation achieves an average relative improvement of 40% in minority-class F1 while incurring only a 16% slowdown in inference throughput, whereas bi-directional message passing incurs substantially higher computational cost. In terms of memory, neighbor-aware aggregation incurs no measurable overhead over PNA (Figure 4).

Figure 4 evaluates scalability in terms of inference throughput and peak GPU memory consumption. Experiments are conducted on the AML Small HI and AML Medium HI datasets. For both training and inference, we use mini-batch neighborhood sampling (Hamilton et al., 2017). We study the effect of neighborhood size, defined as the maximum number of neighbors sampled per hop for each target node in a mini-batch. Importantly, increasing the neighborhood size enlarges the sampled subgraph and increases the complexity.

Figure 4 shows that MEGA-PNA achieves inference throughput comparable to Multi-PNA and higher than Multi-FraudGT, while maintaining peak GPU memory usage similar to Multi-PNA. The figure also includes a variant of MEGA-PNA with bidirectional message passing disabled. This variant attains substantially higher throughput than the full MEGA-PNA model and scales competitively with standard GNN baselines such as PNA in both processing speed and memory efficiency. Importantly, these results, in terms of both throughput rate and memory usage, are consistent with the theoretical analysis presented in Section 3.5.1.

Although all models use two message-passing layers with two-hop sampling and sample at most 100 neighbors per hop for each node, the effective number of sampled neighbors depends on the dataset. For instance, AML Medium HI is denser, with an average node degree approximately $1.5\times$ higher than AML Small HI (see Table 4), providing more neighbors for sampling at each hop. Thus, it yields larger sampled subgraphs, which in turn lead to higher peak GPU memory consumption and lower inference throughput. This reduction cannot be explained solely by neighborhood size or local density; it also reflects the increased cost of sampling from a larger underlying graph. In other words, the size of the input graph indirectly affects throughput. Appendix D provides additional results on training time per epoch and peak GPU memory during training.

## 5 Conclusion

We introduced neighbor-aware aggregation, a novel operator for edge-attributed multigraphs that separates aggregation over multi-edges from aggregation across neighbors. We theoretically established that neighbor-aware aggregation is strictly more expressive than standard single-stage aggregation in edge-attributed multigraph settings. Building on this, we proposed MEGA-GNN, a backbone-agnostic framework that augments standard GNNs to more effectively exploit edge-attributed multigraph structure, while preserving permuta-

tion equivariance and maintaining the same asymptotic complexity as standard GNNs with edge updates. On financial crime benchmarks, MEGA-GNN improved minority-class F1 over baseline GNNs across all dataset variants we evaluated, and over Multi-GNN in 17 of 18 paired comparisons, with the largest absolute and relative gains on the low-illicit AML datasets, where a larger fraction of illicit edges lies on multi-edge node pairs. On temporal user-item interaction datasets, MEGA-GNN also improved ROC-AUC over strong baselines including JODIE and Multi-PNA. These results demonstrate that MEGA-GNN provides a principled and effective approach for learning on edge-attributed multigraphs in which per-neighbor structure carries discriminative information.

**Limitations & Future Work:**  Our theoretical analysis focuses on moment-based aggregators, using the sum and raw moments of orders 2 through $k$. Within this setting, we show that neighbor-aware aggregation is strictly more expressive than single-stage aggregation. Extending this result to other aggregation mechanisms is a natural next step. MEGA-GNN has so far been evaluated on financial transaction datasets and temporal user–item interaction datasets from social platforms. The applicability of MEGA-GNN to broader graph domains, including property graphs with edge attributes, transportation networks (Liu et al., 2020), and cybersecurity datasets (Sarhan et al., 2021; 2022), remains to be explored in future work.

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

# A Limitations of Existing Solutions

## A.1 Multi-Relational GNNs

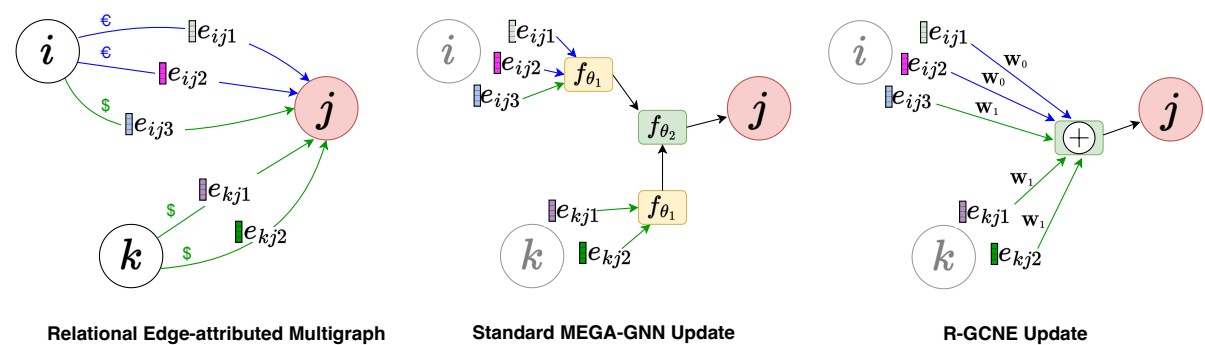

Figure 5: Comparison of MEGA-GNN and R-GCNE architectures. MEGA-GNN performs a neighbor-aware aggregation: it first aggregates multiple edges between the same node pair (edge-level aggregation) and then applies node-level aggregation. In contrast, R-GCNE performs single stage aggregation, and it applies relation-specific transformations by multiplying each edge feature with a learnable weight matrix based on its relation type (currency in the example graph). For clarity, inverse relations and self-loops are omitted in the R-GCNE illustration.

Relational Graph Convolutional Networks (R-GCNs) Schlichtkrull et al. (2018) are specifically designed for *labeled* multigraphs, where each edge is assigned a relation type from a fixed, finite set of discrete labels. R-GCNE (see Appendix C.6) is an extension of R-GCN Schlichtkrull et al. (2018) that uses edge attributes and edge updates. R-GCNE achieves relation-aware message passing by applying distinct learnable weight matrices for each relation type. However, despite this relational specificity, R-GCNE performs a *single-stage aggregation*: messages from all neighbors are summed, optionally scaled by a problem-specific normalization constant. With single-aggregation scheme R-GCNE cannot distinguish between edges originating from the same neighbor (i.e. multi-edges) and edges originating from different neighbors. By contrast, MEGA-GNN introduces a novel neighbor-aware aggregation mechanism. First, it aggregates the attributes of multiple edges connecting the same pair of nodes (edge-level aggregation), capturing intra-pair interactions and edge-specific statistics. Second, the resulting per-neighbor representations are aggregated across distinct neighbors (node-level aggregation). As shown in Theorem 1, this hierarchical design allows MEGA-GNN to compute detailed per-neighbor statistics that standard message-passing GNNs, including R-GCNE, inherently overlook. Such capability is critical in financial transaction networks. A visual comparison of these two architectures is provided in Figure 5, where inverse relations and self-loops are omitted from the R-GCNE diagram for simplicity.

To apply R-GCN to Anti-Money Laundering (AML) datasets in our experiments in Section 4, we converted the multigraph into a multi-relational graph by assigning edge types based on transaction currency. However, this transformation is not always feasible; for example, in the ETH dataset (see Section 4), multigraphs often lack well-defined relation types, limiting the applicability of standard relational GNNs. Importantly, our work is orthogonal to existing relational GNN approaches and naturally extends to multi-relational multigraphs—that is, graphs where multiple edges of the same type exist between the same node pair. This extension enables the development of a hybrid model, MEGA-R-GCNE, which integrates our multi-edge aggregation strategy with R-GCNE-like archi-

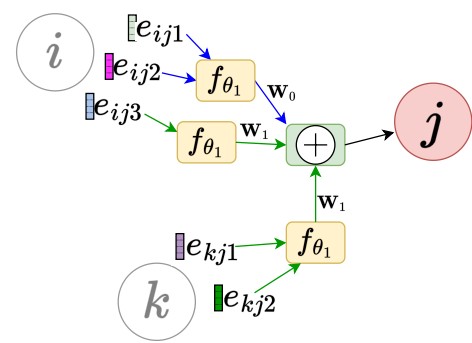

Figure 6: Illustration of MEGA-R-GCNE.

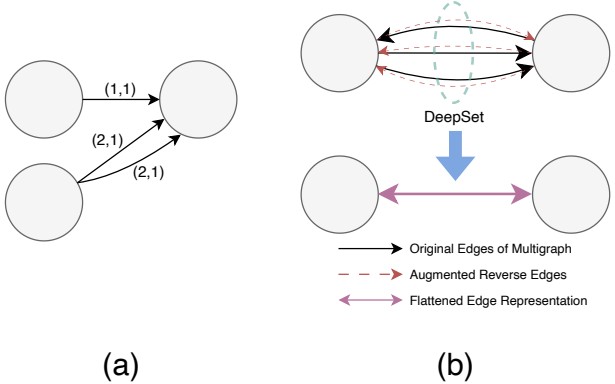

(a)                (b)

Figure 7: (a) Directed multigraph port numbering of Multi-GNN Egressy et al. (2024). (b) Illustration of multigraph to simple graph transformation by ADAMM Sotiropoulos et al. (2023)

tectures, combining the strengths of both approaches. An illustration of the hybrid method is shown in Figure 6, with experimental results presented in Section 4.

## A.2 Multi-GNN, ADAMM and DIAM

Table 8: Related work vs. MEGA-GNN. MP refers to Message Passing, Aggr. stands for Aggregation.

| Features | Multi-GNN | ADAMM | DIAM | MEGA-GNN |
|---|---|---|---|---|
| Bi-directional MP | ✓ | | ✓ | ✓ |
| Edge Embeddings | ✓ | | | ✓ |
| Node Embeddings | ✓ | ✓ | ✓ | ✓ |
| Permutation Equivariance | | ✓ | ✓ | ✓ |
| Neighbor-aware Aggr. | | ✓ | | ✓ |
| Neighbor-aware Aggr. in MP | | | | ✓ |
| Proof of neighbor-aware is more powerful | | | | ✓ |

In the literature, two key works specifically address multigraphs: Multi-GNN Egressy et al. (2024) and ADAMM Sotiropoulos et al. (2023).

**Multi-GNN** introduced a provably powerful GNN architecture for directed multigraphs, incorporating simple adaptations such as reverse message passing, port numbering, and ego IDs You et al. (2021). A notable contribution of Multi-GNN is the multigraph port numbering, which enables the model to distinguish between edges originating from the same neighbor and those from different neighbors (see Figure A.2 (a)). These three adaptations make it possible to assign unique node IDs in connected directed multigraphs, making the Multi-GNN solution universal. However, augmenting edge features with port numbers results in the loss of permutation equivariance (see Proposition 2 and the proof in Appendix B.4). This loss is significant because permutation equivariance is a crucial property for ensuring that the model's predictions remain consistent under arbitrary permutations of nodes or edges in graph learning tasks. Empirical evaluation on the impact of permuting port numbers during inference is presented in Table 4.

**Proposition 2.** *The multigraph port numbering Egressy et al. (2024) is not permutation equivariant.*

The proof of Proposition 2 is given in Appendix B.4.

**ADAMM** aggregates multi-edges between two nodes into a single undirected super-edge (see Figure A.2 (b)), before message passing layers. The initial features for this super-edge are computed using DeepSet Zaheer et al. (2017), incorporating the direction of the edge as an additional edge feature to differentiate between original and augmented reverse edges. The subsequent message passing layers then operate on these

aggregated features. However, this approach loses critical structural information inherent in the multigraph by failing to preserve individual edge features, making it unsuitable for tasks such as edge classification.

Additionally, since ADAMM does not compute latent features for the original edges, it cannot perform multi-edge aggregations repeatedly across multiple message passing layers. Another limitation of ADAMM is its lack of support for bi-directional message passing beyond merely incorporating edge direction as a feature. Previous works have shown that explicit bi-directional message passing improve accuracy for directed multigraphs Egressy et al. (2024).

**DIAM** Ding et al. (2024) models cryptocurrency transaction networks as directed, edge-attributed multigraphs and focuses on learning node representations that capture both temporal transaction patterns and structural discrepancies between illicit and benign accounts. The method introduces Edge2Seq, which constructs sequences from a node's incoming and outgoing edges separately and encodes them with GRUs to produce node-level embeddings. Although this sequence-based processing yields informative node-level representations, it does not update edge representations, as all edge information is immediately merged into node embeddings. DIAM further employs a Multigraph Discrepancy (MGD) module that performs directed message passing using both neighbor features and their differences from the target node, a design meant to emphasize behavioral deviations of illicit nodes rather than rely on homophily.

Despite achieving strong results on standard node classification benchmarks like ETH (see Table 3), the original DIAM model only computes node embeddings and does not support edge classification. However, we have extended it to include an edge classification head that leverages DIAM's node embeddings and the original edge features, which enabled us to apply DIAM also to AML datasets (Altman et al., 2023) which we include the results to Appendix E. The results in Table 19 show that on the AML task, DIAM outperformed by our method (MEGA-PNA), clearly reflecting the importance of our edge-attributed multigraph modeling approach in effectively capturing complex transaction patterns.

# B  Proofs

## B.1  Proof of Theorem 1

For any function $f : \mathcal{X} \to \mathcal{Y}$, we write $\mathrm{Im}(f)$ to denote its image, i.e., the set $\{f(x) \mid x \in \mathcal{X}\} \subseteq \mathcal{Y}$.

*Proof.* We assume that each edge feature vector is equipped with a constant, i.e., $1 \in \mathbb{R}$. Hence, we can note that by using $g_1$, the model can compute the cardinality of the multiset.

We prove the claim by establishing two parts: (i) $\mathrm{Im}(\mathcal{T}_{\text{single-stage}}) \subseteq \mathrm{Im}(\mathcal{T}_{\text{neighb-aware}})$, (ii) $\mathrm{Im}(\mathcal{T}_{\text{single-stage}}) \neq \mathrm{Im}(\mathcal{T}_{\text{neighb-aware}})$.

**Part (i)**  As stated in the Definition 3, $\mathcal{T}_{\text{single-stage}}$ aggregates all edge features in the neighborhood $\mathcal{X}_j$, treating the neighborhood as a single multiset,

$$\mathcal{X}_{\text{flat}} := \biguplus_{X_{ij} \in \mathcal{X}_j} X_{ij}.$$

Then the output of the single-stage aggregation is given by,

$$\mathcal{T}_{\text{single-stage}}(\mathcal{X}_j) = f_\theta(\mathcal{X}_{\text{flat}}) = \mathrm{MLP}_\theta\left([g_1(\mathcal{X}_{\text{flat}}) \| \ldots \| g_k(\mathcal{X}_{\text{flat}})]\right).$$

Neighbor-aware aggregation scheme first applies $f_{\theta_1}$ to each multiset $X_{ij}$ *separately*, then $f_{\theta_2}$ is applied to resulting multiset of vectors in the second stage:

$$f_{\theta_1}(X_{ij}) = \mathrm{MLP}_{\theta_1}\left([g_1(X_{ij}) \| \ldots \| g_k(X_{ij})]\right),$$

and

$$\mathcal{T}_{\text{neighb-aware}}(\mathcal{X}_j) = f_{\theta_2}\left(\{\{f_{\theta_1}(X_{ij}) \mid X_{ij} \in \mathcal{X}_j\}\}\right).$$

Now we show that $\mathcal{T}_{\text{neighb-aware}}$ can compute what $\mathcal{T}_{\text{single-stage}}$ can compute.

We first consider the case $r = 1$, corresponding to the sum aggregator:

$$g_1(\mathcal{X}_{\text{flat}}) = \sum_{i \in N_{\text{in}}(j)} g_1(X_{ij})$$

Thus, by applying $g_1$ in both stages, the model can compute $g_1(\mathcal{X}_{\text{flat}})$.

Now for $2 \leq r \leq k$, the raw moments of the flattened multiset $\mathcal{X}_{\text{flat}}$, can be expressed as a weighted average of the raw moments of the individual multisets $X_{ij}$:

$$g_r(\mathcal{X}_{\text{flat}}) = \frac{1}{n} \sum_{X_{ij} \in \mathcal{X}_j} P_{ij} \cdot g_r(X_{ij}), \quad 2 \leq r \leq k,$$

where $n := \sum_i P_{ij}$ is the total number of edges.

Since each edge feature is equipped with a constant 1, the cardinality $P_{ij} = |X_{ij}|$ can be computed using the sum aggregator $g_1$. Thus, $f_{\theta_1}$ can compute $P_{ij} \cdot g_r(X_{ij})$ for each $r$.

In the second stage, the multiset $\{\{f_{\theta_1}(X_{ij}) \mid X_{ij} \in \mathcal{X}_j\}\}$ contains all such terms, and the sum aggregator $g_1$ in $f_{\theta_2}$ can compute

$$\sum_{X_{ij} \in \mathcal{X}_j} P_{ij} \cdot g_r(X_{ij}), \quad n = \sum_i P_{ij}$$

thus enabling $f_{\theta_2}$ to compute the raw moments $g_r(\mathcal{X}_{\text{flat}})$.

Hence, we conclude that:

$$\text{Im}(\mathcal{T}_{\text{single-stage}}) \subseteq \text{Im}(\mathcal{T}_{\text{neighb-aware}}).$$

**Part (ii):** We now show that $\text{Im}(\mathcal{T}_{\text{single-stage}}) \neq \text{Im}(\mathcal{T}_{\text{neighb-aware}})$.

We begin by defining a simple function that cannot be computed by any single-stage aggregation scheme.

$$F_r(\mathcal{X}_j) := \sum_{X_{ij} \in \mathcal{X}_j} g_r(X_{ij}), \quad 2 \leq r \leq k$$

Such functions, $F_r$, are computable by $\mathcal{T}_{\text{neighb-aware}}$ since by design it preserves the partitioning over distinct neighbors of node $j$, allowing $f_{\theta_2}$ to operate on a multiset of per-neighbor representations.

In contrast, any function computed by $\mathcal{T}_{\text{single-stage}}$ is in the form:

$$\mathcal{T}_{\text{single-stage}}(\mathcal{X}_j) = f_\theta(\mathcal{X}_{\text{flat}}) = \text{MLP}_\theta\left([g_1(\mathcal{X}_{\text{flat}}) \| \ldots \| g_k(\mathcal{X}_{\text{flat}})]\right).$$

Let $n := |\mathcal{X}_{\text{flat}}|$, we now analyze two distinct cases:

**Case 1:** $k \geq n$ When $k \geq n$, the number of aggregators in $f_\theta$ is sufficient to reconstruct the entire neighborhood multiset $\mathcal{X}_{\text{flat}}$ without loss of information, as established in Theorem 1 of Corso et al. (2020). However, since $\mathcal{X}_{\text{flat}}$ does not contain any information about which neighbor $i$ each edge originates from, even a full reconstruction of the neighborhood does not allow the recovery of per-neighbor partitions. Therefore, the function $F_r$ is incomputable by $\mathcal{T}_{\text{single-stage}}$.

**Case 2:** $k < n$ In this case $\mathcal{T}_{\text{single-stage}}$ cannot even reconstruct the full multiset $\mathcal{X}_{\text{flat}}$, as the number of aggregators $k$ is insufficient to discriminate between multisets of size $n$, as stated in Theorem 1 in Corso et al. (2020). As a result, the function $F_r$ remains incomputable by $\mathcal{T}_{\text{single-stage}}$.

Therefore, we have proven that neighbor-aware aggregation induces a strictly larger image than single-stage aggregation,

$$\text{Im}(\mathcal{T}_{\text{single-stage}}) \subsetneq \text{Im}(\mathcal{T}_{\text{neighb-aware}}).$$

$\square$

## B.2 Proof of Theorem 3

*Proof.* Let $d$ denote the node and edge embedding dimension. Following prior work (e.g., Chiang et al. (2019); Li et al. (2021a)), we analyze activation memory and exclude static graph-structure tensors (e.g., adjacency/edge-index), since activation memory is typically the dominant bottleneck.

In standard message-passing GNNs with edge features, storing node and edge activations requires

$$\mathcal{O}(|\mathcal{V}|d + |\mathcal{E}|d).$$

The only additional component introduced by MEGA-GNN is an index tensor of size $\mathcal{O}(|\mathcal{E}|)$ to identify multi-edges. Hence, the total memory becomes $\mathcal{O}(|\mathcal{V}|d + |\mathcal{E}|d + |\mathcal{E}|)$. Since $d \geq 1$, this simplifies to

$$\mathcal{O}(|\mathcal{V}|d + |\mathcal{E}|d).$$

$\square$

## B.3 Proof of Proposition 1

We adopt the notation and terminology introduced in Section 3.1 of this paper to ensure consistency and ease of reference.

*Proof:* The proposed message passing layer performs two aggregations over the neighborhood of a target node $j$. The first is the multi-edge aggregation, in which the latent features of the multi-edges are aggregated at artificial nodes. The multiset of such features are denoted as,

$$X_{ij} = \{\{\mathbf{e}_{ijp} \mid p \in [P_{ij}]\}\} \tag{12}$$

The vectors in the multiset $X_{ij}$ are aggregated on artificial nodes,

$$\mathbf{h}_{ij} = f_{\theta_1}(X_{ij}). \tag{13}$$

Since aggregators in $f_{\theta_1}$ are assumed to be permutation invariant, for any permutation function $\rho$ acting on multi-edges, we have $f_{\theta_1}(\rho \cdot X_{ij}) = f_{\theta_1}(X_{ij})$.

The second aggregation is then performed over the neighborhood of the target nodes, all of which happen to be artificial nodes associated with distinct neighbors in the original graph (see Figure 2).

$$H_{N_{in}(j)} = \{\{\mathbf{h}_{ij} \mid (i,j) \in N_{in}(j))\}\} \in \mathcal{M}_d. \tag{14}$$

Hence, the second aggregation operates over the multiset $H_{N_{in}(j)}$,

$$\mathbf{x}_j = f_{\theta_2}(H_{N_{in}(j)}) \in \mathcal{M}_d. \tag{15}$$

Again since the aggregators in $f_{\theta_2}$ are assumed to be permutation invariant for any permutation function $\pi$ acting on the neighbors of a target node $j$, we have $f_{\theta_2}(\pi \cdot H_{N_{in}(j)}) = f_{\theta_2}(H_{N_{in}(j)})$.

Our framework MEGA-GNN, integrates the neighbor-aware aggregation scheme (as defined in Definition 4) using aggregation functions $f_{\theta_1}$ and $f_{\theta_2}$ within a single message passing layer, as detailed in Section 3.3. Since the composition of permutation invariant functions remains permutation invariant, our message passing layer $(f_{\theta_1} \circ f_{\theta_2})$ is invariant to the permutations of neighboring nodes and edges. Unlike simple graphs, node permutations do not directly imply edge permutations in multigraphs due to the presence of multi-edges. Thus, we explicitly define the permutation of multi-edges, $\rho$, ensuring that our message passing layer remains permutation-invariant to both nodes and edges in the neighborhood of the target node.

Finally, as demonstrated by Bronstein et al. (2021), the composition of permutation invariant layers $(f = f_{\theta_1} \circ f_{\theta_2} \circ f_{\theta_1} \circ f_{\theta_2} \cdots)$ allows the construction of functions $f$ that are equivariant to symmetry group actions. In the multigraph domain, this symmetry group includes permutations of both nodes and edges. The overall permutation equivariance of the MEGA-GNN model follows from the fact that each permutation invariant message passing layer operates independently on each node's neighborhood, regardless of the ordering of nodes or edges. Specifically, for any permutation $g \in \sum_n$ acting on the set of node and edges, the model's output satisfies $f(g \cdot X) = g \cdot f(X)$.

## B.4 Proof of Proposition 2

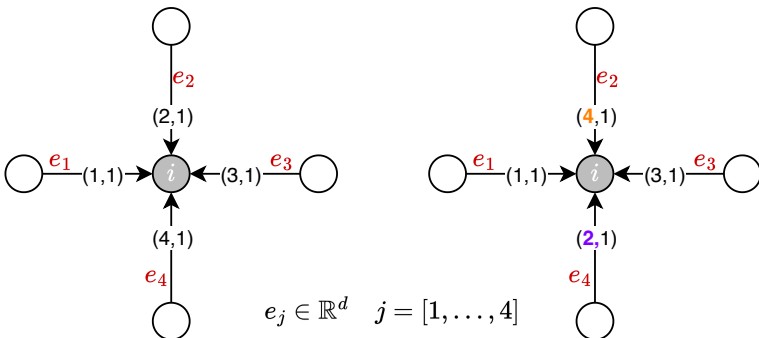

Figure 8: Illustration of counter example for the permutation equivariance of Multi-GNN Egressy et al. (2024). The left panel shows the graph $\mathcal{G}$ with one permutation of the port numbering, while the right panel illustrates a different permutation of the assigned port numbers. Assume each edge has distinct features, i.e $e_2 \neq e_4$.

For consistency we use the same notation introduced in Section 3.1. Let $\mathcal{G} = (\mathcal{V}, \mathcal{E})$ be a multigraph with node features $\mathbf{x}_i \in \mathbb{R}^D$ and edge features $\mathbf{e}_{ijp} \in \mathbb{R}^K$. We assume that each edge carries a distinct feature vector. Each edge $e \in \mathcal{E}$ is assigned a port number $\rho(e)$ by a given port numbering scheme, and these port numbers are incorporated into the edge features, as proposed by Egressy et al. (2024).

The assignment of port numbers is arbitrary. For a node $i$ with $d$ incoming edges, there are $d!$ possible port numbering, corresponding to all possible permutations.

The latent feature of the node $i$ at layer $l$ is computed as:

$$\mathbf{x}_i^{(l)} = \sum_{j \in N(i)} \sum_{p \in P_{ji}} \phi\big(\mathbf{x}_j^{(l-1)}, [\mathbf{e}_{jip}^{(l-1)} \mid\mid \rho(j, i)]\big). \tag{16}$$

where $\phi$ is the message function, and $\rho(j, i) \in \mathbb{R}^2$ is the port numbers assigned to the edge between node $j$ and node $i$, as shown in Figure A.2(a)

We proceed by proof by contradiction. Suppose that the GNN with port numbering is permutation equivariant at the graph level, that is, permuting the node and edge indices results in an equivalent permutation of the output given as in Definition 5. This property requires the model to be permutation invariant over each node's neighborhood: reordering the incoming edges (i.e., permuting the port numbers) should not affect a node's representation.

Let $\sigma$ be a permutation of the port numbers. Applying this permutation to the port numbers of the edges yields a new port assignment $\rho_\sigma(e)$. The updated representation of node $i$ at layer $l$ under this permuted port assignment is:

$$\hat{\mathbf{x}}_i^{(l)} = \sum_{j \in N(i)} \sum_{p \in P_{ji}} \phi\big(\mathbf{x}_j^{(l-1)}, [\mathbf{e}_{jip}^{(l-1)} \mid\mid \rho_\sigma(j, i)]\big). \tag{17}$$

By assumption:

$$\mathbf{x}_i^{(l)}(\rho) = \hat{\mathbf{x}}_i^{(l)}(\rho_\sigma) \tag{18}$$

However, since $\phi$ explicitly depends on the port number $\rho(j, i)$, permuting the port numbers alters the input to $\phi$ via concatenated feature $[\mathbf{e}_{jip}^{(l-1)} \mid\mid \rho_\sigma(j, i)]$. Assuming that each edge carries a distinct feature vector, as is typical in settings like financial transaction networks, this change affects the resulting messages

and, consequently, the updated representation of node $i$. Hence permuting assigned port numbers leads to $\mathbf{x}_i^{(l)}(\rho) \neq \hat{\mathbf{x}}_i^{(l)}(\rho_\sigma)$, violating permutation invariance over the neighborhood of $i$, contradicting the assumption of GNN being permutation equivariant.

Figure 8 illustrates this contradiction: node $i$ has four incoming edges, and permuting their port numbers leads to different messages and a different update. We thus conclude that arbitrary port numbering breaks permutation equivariance.

## C   Implementation Details

To support reproducibility, the full source code, training scripts, dataset preprocessing pipeline, and all experiment configuration files will be made publicly available upon publication.

### C.1   Architecture Diagrams

Figure 2 illustrate a single layer of the MEGA-GNN architecture. Figure 9 shows the MEGA-GNN layer equipped with bi-directional message-passing capabilities. The figures use the same notation as in Sections 3.3 and 3.4 for clarity.

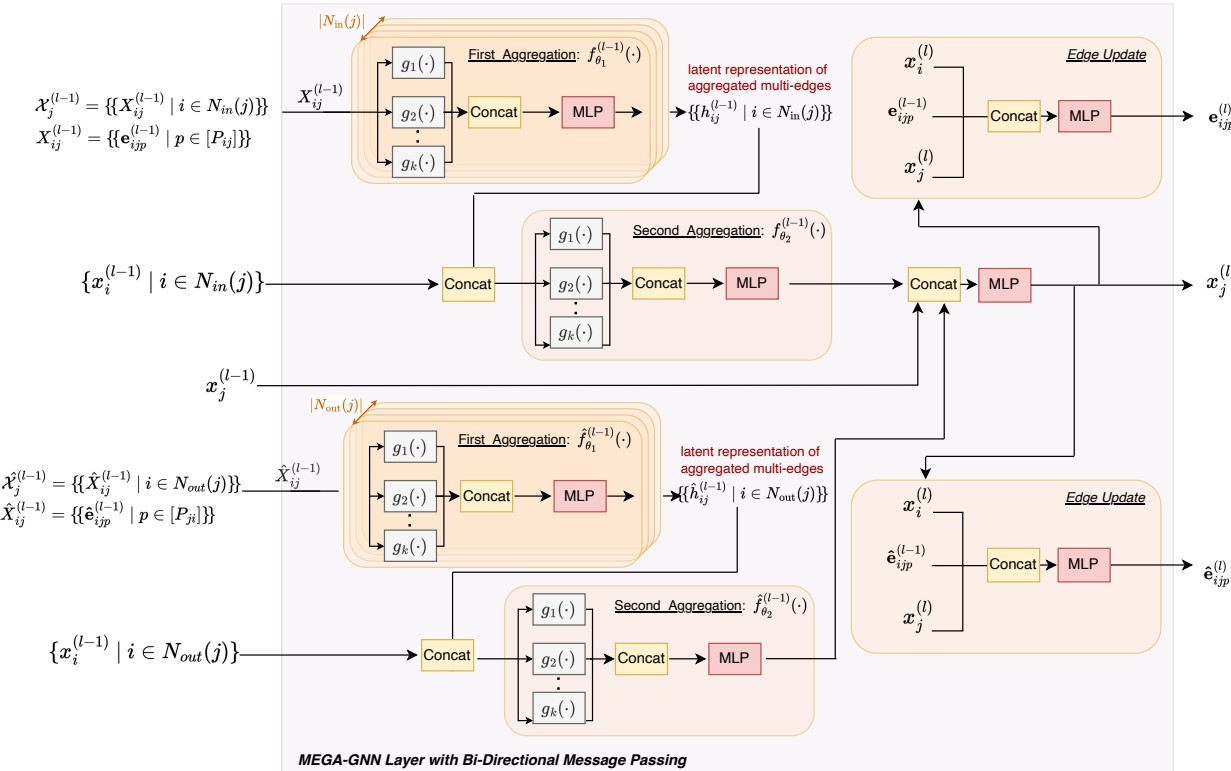

Figure 9: Overview of the MEGA-GNN layer with a bi-directional message passing. In directed multigraphs, reverse edges are added opposite to the original edges. Separate message computations are performed for original and reversed edges. The diagram illustrates the message passing scheme described in Sections 3.3 and 3.4, using consistent notation.

### C.2   Comparison points

Table 9 summarizes the baseline methods and multigraph adaptations evaluated in our experiments.

Table 9: Backbone architectures and multigraph adaptations: checkmarks indicate which combinations are evaluated in our experiments.

| Multigraph Adaptations | GIN | PNA | GenAgg | R-GCN | FraudGT |
|---|---|---|---|---|---|
| Base (no adaptations) | ✓ | ✓ | ✓ | ✓ | ✓ |
| Multi Egressy et al. (2024) | ✓ | ✓ | ✓ | | ✓ |
| ADAMM Sotiropoulos et al. (2023) | ✓ | ✓ | ✓ | | |
| MEGA (ours) | ✓ | ✓ | ✓ | ✓ | |

## C.3 Hyperparameters

Table 10: Hyperparameter settings for AML and ETH datasets

| | GIN | | PNA | | R-GCNE |
|---|---|---|---|---|---|
| | AML | ETH | AML | ETH | AML |
| $lr$ | 0.003 | 0.006 | 0.0008 | 0.0008 | 0.003 |
| $hidden\_dim$ | 64 | 32 | 20 | 20 | 32 |
| $batch\_size$ | 8192 | 4096 | 8192 | 4096 | 8192 |
| $dropout$ | 0.1 | 0.1 | 0.28 | 0.1 | 0.1 |
| $w\_ce1, w\_ce1$ | 1, 6.27 | 1, 6.27 | 1, 7 | 1, 3 | 1, 6.27 |

For each base GNN model and dataset, we utilized a distinct set of hyperparameters, as detailed in Table 10. The MEGA-GenAgg and Multi-GenAgg models employed the aggregation function proposed by Kortvelesy et al. (2023). In all experiments involving GenAgg, we adopted the default layer sizes of $(1, 2, 2, 4)$, and both the $a$ and $b$ parameters were made learnable, allowing the model to tailor the aggregation function to the specific downstream task. Additionally, for the GenAgg experiments, we applied the hyperparameters configured for GIN-based models as shown in Table 10.

For the AML dataset, the model was operated on neighborhoods constructed around the seed edges, while for the ETH dataset, the neighborhoods were selected around the seed nodes. In both datasets, we sampled 2-hop neighborhoods, selecting 100 neighbors per hop.

## C.4 Dataset Statistics and Splits

Table 11: Statistics of JODIE datasets.

| Dataset | # Nodes | # Edges | Illicit Rate | Split [%] |
|---|---|---|---|---|
| MOOC | 7,144 | 411,749 | 0.98% | 60/20/20 |
| Reddit | 10,984 | 672,447 | 0.05% | 60/20/20 |
| Wikipedi | 9,227 | 157,474 | 0.14% | 60/20/20 |

**Jodie Data Split:** We adopt the same temporal splitting strategy as proposed in Kumar et al. (2019), which follows train-validation-test split based on timestamps defined on the edges. Table 11 shows the dataset details.

**AML Data Split:** We adopt the same temporal splitting strategy as proposed in Egressy et al. (2024), which follows train-validation-test split based on transaction timestamps. Specifically, we sort all transactions and define two cut-off points, $t_1$ and $t_2$, to partition the data as summarized in Table 12. Transactions occurring before $t_1$ are used for training, those between $t_1$ and $t_2$ for validation, and those after $t_2$ for testing. Since validation and test transactions may depend on patterns in earlier activity, we construct three

Table 12: Statistics of AML, ETH datasets.

| Dataset | # Nodes | # Edges | Illicit Rate | Split [%] |
|---|---|---|---|---|
| AML Small HI | 515,088 | 5,078,345 | 0.102% | 64/19/17 |
| AML Small LI | 705,907 | 6,924,049 | 0.051% | 64/19/17 |
| AML Medium HI | 2,077,023 | 31,898,238 | 0.110% | 61/17/22 |
| AML Medium LI | 2,032,095 | 31,251,483 | 0.051% | 61/17/22 |
| AML Large HI | 2,116,168 | 179,702,229 | 0.124% | 60/20/20 |
| AML Large LI | 2,070,980 | 176,066,557 | 0.057% | 60/20/20 |
| ETH | 2,973,489 | 13,551,303 | 0.04% | 65/15/20 |

Table 13: Detailed statistics of AML datasets.

| Statistic | Small | | Medium | | Large | |
|---|---|---|---|---|---|---|
| | LI | HI | LI | HI | LI | HI |
| **Graph scale** | | | | | | |
| Nodes | 705,907 | 515,088 | 2,032,095 | 2,077,023 | 2,070,980 | 2,116,168 |
| Edges | 6,924,049 | 5,078,345 | 31,251,483 | 31,898,238 | 176,066,557 | 179,702,229 |
| **Degree (in)** | | | | | | |
| Mean in-degree | 9.81 | 9.86 | 15.38 | 15.36 | 85.02 | 84.92 |
| Std. in-degree | 13.00 | 12.97 | 21.67 | 21.64 | 123.69 | 123.52 |
| Max in-degree | 1,553 | 1,084 | 6,811 | 6,842 | 31,762 | 31,906 |
| **Degree (out)** | | | | | | |
| Std. out-degree | 325.51 | 285.15 | 926.65 | 918.93 | 5,314.04 | 5,270.29 |
| Max out-degree | 222,037 | 168,672 | 1,092,870 | 1,076,979 | 6,327,667 | 6,233,962 |
| **Multi-edge structure** | | | | | | |
| Unique (src, dst) pairs | 1,384,862 | 1,015,736 | 4,363,197 | 4,476,959 | 8,177,994 | 8,466,789 |
| Mean multi-edge multiplicity | 5.00 | 5.00 | 7.16 | 7.12 | 21.53 | 21.22 |
| Max multi-edge multiplicity | 81 | 89 | 145 | 118 | 877 | 690 |
| Std. multi-edge multiplicity | 7.38 | 7.39 | 12.00 | 11.96 | 56.38 | 55.95 |
| Median multi-edge multiplicity | 3 | 3 | 4 | 4 | 8 | 8 |
| Frac. edges on pairs with $\geq 2$ edges | 0.9109 | 0.9106 | 0.9346 | 0.9341 | 0.9791 | 0.9785 |
| Frac. pairs with $\geq 2$ edges | 0.5547 | 0.5529 | 0.5318 | 0.5307 | 0.5502 | 0.5447 |
| **Illicit placement** | | | | | | |
| Mean edge multiplicity (pairs with $\geq 1$ illicit edge) | 5.18 | 3.74 | 8.64 | 5.10 | 44.63 | 24.29 |
| Max edge multiplicity (pairs with $\geq 1$ illicit edge) | 51 | 41 | 84 | 84 | 501 | 501 |
| Std. edge multiplicity (pairs with $\geq 1$ illicit edge) | 8.02 | 6.64 | 13.84 | 10.29 | 81.76 | 60.81 |
| Frac. illicit edges on pairs with multiplicity $= 1$ | 0.6168 | 0.7292 | 0.5583 | 0.6856 | 0.2963 | 0.5424 |
| Frac. illicit edges on pairs with multiplicity $\geq 2$ | 0.3832 | 0.2708 | 0.4417 | 0.3144 | 0.7037 | 0.4576 |

dynamic graph snapshots at times $t_1$, $t_2$, and $t_3 = t_{\max}$, the latest timestamp in the dataset. The train graph includes only training transactions and their corresponding nodes. The validation graph includes both training and validation transactions but computes metrics only on the validation indices. Similarly, the test graph contains all transactions in the given dataset, with evaluation performed solely on the test indices. This dynamic setup mirrors real-world usage in financial institutions, where systems must detect anomalies in new batches of transactions while leveraging historical context.

**ETH Data Split:** Similar to AML we use a temporal train-validation-test split. We order the nodes by the first transaction they are involved in (either as sender or receiver) before splitting. Again this gives us threshold times t1 and t2, and we use these times to create our train, validation, and test graphs.

## C.5 MEGA Variants

In this section, we provide detailed descriptions of the MEGA-GIN, MEGA-PNA, MEGA-GenAgg, and MEGA-RGCNE models used in our study.

As introduced in Sections 3.1 and 3.2, our framework employs a neighbor-aware aggregation scheme with two aggregation functions, $f_{\theta_1}$ and $f_{\theta_2}$, each constructed from a set of $k$ aggregators $g_1, \ldots, g_k : \mathcal{M}_d \to \mathbb{R}^d$. Specifically, we define:

$$f_\theta(X) := \mathrm{MLP}_\theta\big([g_1(X) \,\|\, \ldots \,\|\, g_k(X)]\big), \quad f_\theta : \mathcal{M}_d \to \mathbb{R}^{d'}. \tag{19}$$

The MEGA variants differ in the choice and number of aggregators used in the two-stage process.

**MEGA-GIN.** Following the Graph Isomorphism Network (GIN) model proposed by Xu et al.Xu et al. (2019), MEGA-GIN uses a single aggregator, namely SUM, in both $f_{\theta_1}$ and $f_{\theta_2}$. That is, $k = 1$ and $g_1 = $ SUM.

**MEGA-PNA.** The MEGA-PNA builds on the Principal Neighbourhood Aggregation (PNA) framework proposed by Corso et al. Corso et al. (2020), which combines multiple statistical aggregators. Accordingly, we use $k = 4$ aggregators: MEAN, MAX, MIN, and STD, applied in both $f_{\theta_1}$ and $f_{\theta_2}$.

**MEGA-GenAgg.** The MEGA-GenAgg employs a single, learnable aggregator as introduced in the GenAgg framework Kortvelesy et al. (2023). Unlike fixed statistical functions, the aggregator in GenAgg is parameterized and trained end-to-end. Accordingly, we set $k = 1$, and use $g_1 = $ GENAGG in both $f_{\theta_1}$ and $f_{\theta_2}$.

**MEGA-RGCNE.** The MEGA-RGCNE variant integrates the expressive multi-aggregator scheme of PNA Corso et al. (2020) in the first aggregation stage $f_{\theta_1}$, where we use $k = 4$ aggregators: MEAN, MAX, MIN, and STD. In the second stage $f_{\theta_2}$, we incorporate relation-specific transformation matrices $W_r^{(l)}$, applying distinct linear transformations for each relation type $r$, as shown in Figure 6. This design demonstrates the flexibility of the MEGA framework, which allows different aggregation strategies to be combined across stages.

## C.6 Edge-Augmented R-GCN (R-GCNE)

To incorporate edge attributes, we extend the standard Relational Graph Convolutional Network (R-GCN) Schlichtkrull et al. (2018) by introducing the Relational Graph Convolutional Network with Edge features (R-GCNE). In this variant, edge features are also included in the message passing formulation.

Consider the notation introduced in Section 3.1. Building on that, we define the set of relation types $\mathcal{R}$, where each edge $(j, r, i) \in \mathcal{E}$ is labeled with a relation $r \in \mathcal{R}$. Each relation type $r$ is associated with a learnable transformation matrix $W_r^{(l)} \in \mathbb{R}^{d \times d}$, and we define $W_0^{(l)} \in \mathbb{R}^{d \times d}$ as the transformation matrix applied to a node's own features (i.e., self-loop) at layer $l$. For each node $i \in \mathcal{V}$, we define the relation-specific neighborhood as $N_i^r := \{j \in \mathcal{V} \mid (j, r, i) \in \mathcal{E}\}$, representing the set of incoming neighbors connected via relation $r$.

The message passing equations for both models are defined as follows:

- **R-GCN Message Passing Equation:**

$$x_i^{(l+1)} := \sigma\left(x_i^{(l)} W_0^{(l)} + \sum_{r \in \mathcal{R}} \sum_{j \in N_i^r} \frac{1}{|N_i^r|} x_j^{(l)} W_r^{(l)}\right), \quad x_i^{(l+1)} \in \mathbb{R}^d.$$

- **R-GCNE Message Passing Equation:**

$$x_i^{(l+1)} := \sigma\left(x_i^{(l)} W_0^{(l)} + \sum_{r \in \mathcal{R}} \sum_{j \in N_i^r} \frac{1}{|N_i^r|} \left(x_j^{(l)} W_r^{(l)} + e_{ji}^{(l)} W_r^{(l)}\right)\right), \quad x_i^{(l+1)} \in \mathbb{R}^d.$$

If R-GCNE is applied to multigraphs with edge relations, we must also account for multiple edges between the same pair of nodes with the same relation. To handle this, similar to Section 3.1, we define the multiset

$$X_{ij,r} = \{\{\mathbf{e}_{ijp,r} \mid p \in [P_{ij,r}]\}\}$$

denote the multiset of edge feature vectors from node $j$ to node $i$ under relation $r$, where $P_{ij,r}$ is the number of such edges. We denote the feature of the $p$-th such edge as $e_{jip,r}^{(l)} \in \mathbb{R}^d$.

Then the formulation of R-GCNE on multigraphs with relation types becomes:

$$x_i^{(l+1)} := \sigma \left( x_i^{(l)} W_0^{(l)} + \sum_{r \in R} \sum_{j \in N_i^r} \sum_{p=1}^{P_{jri}} \frac{1}{|N_i^r|} (x_j^{(l)} W_r^{(l)} + e_{jip,r}^{(l)} W_r^{(l)}) \right) \in \mathbb{R}^d.$$

## D Computation and Memory Costs

Table 14: Training time (seconds per epoch) and peak GPU memory usage on the AML Small HI dataset. All models follow the experimental setup of Section 4.3.1; MEGA and Multi variants include bi-directional message passing and ego IDs. K denotes $10^3$

| Model | # Params | Train sec/ep | Peak GPU Mem. Usage (GB) |
|---|---|---|---|
| GIN | 69.6K | 2.17 | 0.52 |
| PNA | 32.2K | 2.06 | 0.29 |
| GenAgg | 69.7K | 6.02 | 1.13 |
| FraudGT | 182.4K | 27.51 | 1.62 |
| R-GCNE | 47.7K | 10.70 | 1.32 |
| Multi-GIN | 128.3K | 8.73 | 2.18 |
| Multi-PNA | 60.0K | 7.46 | 1.33 |
| Multi-GenAgg | 128.4K | 23.97 | 5.75 |
| Multi-FraudGT | 243.7K | 92.36 | 8.26 |
| MEGA-GIN | 161.3K | 9.14 | 1.95 |
| MEGA-PNA | 79.2K | 7.83 | 1.44 |
| MEGA-GenAgg | 128.4K | 28.38 | 6.95 |
| MEGA-R-GCNE | 138.3K | 17.43 | 2.48 |

Table 14 reports the number of parameters, training time per epoch, and peak GPU memory usage for all models on AML Small HI. Training time is averaged over 256 iterations per epoch. All experiments were conducted on a single machine equipped with an NVIDIA GeForce RTX 4090 (24 GB), an AMD Ryzen 9 7950X CPU, and 64 GB RAM, using PyTorch 2.2.2 and PyTorch Geometric 2.5.2 with CUDA 11.8.

The results reflect a clear trade-off between efficiency and model capacity. Simpler architectures such as GIN and PNA are both faster and more memory-efficient. In contrast, multigraph extensions (Multi and MEGA) introduce additional computational overhead, leading to increased runtime and higher memory consumption. As shown in Table 15, this additional cost is accompanied by consistent gains in minority-class F1 across AML benchmarks. The higher parameter count of GIN compared to PNA in our setup is due to the selected hyperparameters (Table 10), particularly the larger hidden dimension used for GIN.

## E Additional Experiments and Analyses

### E.1 Full Results and Additional Metrics on AML

Table 15 reports complete minority-class F1 scores. We also provide complementary Precision and Recall tables (Tables 16 and 17) for the AML edge classification task.

Table 15: Minority class F1 scores (%) for six AML datasets using different GNN baselines (GIN,PNA, GenAgg, FraudGT and R-GCNE) and multigraph adaptations (Multi and MEGA). We extended R-GCNE to support only MEGA adaptations. Avg Rank denotes the average rank of each model across all six datasets, where rank 1 corresponds to the best performing model on a given dataset (lower is better).

| Model | Small HI | Small LI | Medium HI | Medium LI | Large HI | Large LI | Rank ($\downarrow$) |
|---|---|---|---|---|---|---|---|
| GFP+LightGBM (Blanuša et al., 2024) | 62.86 ± 0.25 | 20.83 ± 1.50 | 59.48 ± 0.15 | 20.85 ± 0.38 | 48.67 ± 0.24 | 17.09 ± 0.46 | 12.83 |
| GFP+XGBoost (Blanuša et al., 2024) | 63.23 ± 0.17 | 27.30 ± 0.33 | 65.70 ± 0.26 | 28.16 ± 0.14 | 42.68 ± 12.93 | 24.23 ± 0.12 | 10.83 |
| GIN | 46.50 ± 4.11 | 19.93 ± 3.55 | 58.65 ± 2.50 | 25.36 ± 1.49 | 49.80 ± 1.38 | 4.99 ± 3.66 | 14.00 |
| PNA | 62.96 ± 1.43 | 21.02 ± 4.05 | 66.87 ± 1.87 | 31.79 ± 2.30 | 55.01 ± 1.94 | 20.47 ± 1.93 | 10.00 |
| GenAgg | 56.45 ± 2.94 | 21.03 ± 2.23 | 54.21 ± 7.90 | 20.72 ± 2.60 | 52.23 ± 4.29 | 9.23 ± 3.07 | 13.67 |
| R-GCNE | 63.91 ± 3.18 | 37.40 ± 1.61 | 65.71 ± 0.61 | 35.70 ± 0.99 | 58.26 ± 1.08 | 23.32 ± 0.73 | 8.33 |
| FraudGT | 69.68 ± 1.58 | 28.69 ± 2.05 | 63.38 ± 0.87 | 24.02 ± 0.52 | 54.35 ± 1.65 | 11.02 ± 2.65 | 10.67 |
| Multi-GIN | 62.66 ± 1.73 | 32.21 ± 0.99 | 67.72 ± 0.94 | 31.24 ± 2.12 | 71.44 ± 1.25 | 9.46 ± 8.85 | 9.67 |
| Multi-PNA | 67.35 ± 2.89 | 35.39 ± 3.93 | 76.12 ± 0.69 | 43.81 ± 0.51 | 72.35 ± 1.14 | 33.54 ± 2.04 | 5.17 |
| Multi-GenAgg | 64.92 ± 3.85 | 36.36 ± 4.07 | 66.45 ± 1.30 | 37.72 ± 0.73 | 71.70 ± 0.77 | 32.18 ± 1.01 | 6.83 |
| Multi-FraudGT | **75.81 ± 0.75** | _45.69 ± 1.14_ | 75.97 ± 0.18 | 44.66 ± 0.58 | 73.04 ± 0.59 | _35.49 ± 0.52_ | 2.83 |
| MEGA-GIN (ours) | 70.83 ± 2.19 | 43.67 ± 0.55 | 70.77 ± 2.76 | _46.05 ± 1.64_ | 70.41 ± 2.74 | 11.64 ± 1.64 | 5.67 |
| MEGA-PNA (ours) | 74.01 ± 1.55 | **46.32 ± 2.07** | **78.26 ± 0.11** | **49.40 ± 0.54** | **76.95 ± 0.44** | **38.31 ± 1.53** | **1.33** |
| MEGA-GenAgg (ours) | _74.48 ± 0.84_ | 46.30 ± 0.42 | _76.70 ± 0.32_ | 44.90 ± 0.06 | _73.58 ± 0.97_ | 34.84 ± 0.57 | _2.33_ |
| MEGA-RGCNE (ours) | 70.65 ± 1.80 | 40.92 ± 2.69 | 74.48 ± 0.25 | 41.21 ± 0.45 | 67.41 ± 3.38 | 27.25 ± 0.82 | 5.83 |

Table 16: Precision scores (%) on AML edge classification task. Best result is indicated with **bold**.

| Model | Small HI | Small LI | Medium HI | Medium LI | Large HI | Large LI |
|---|---|---|---|---|---|---|
| GIN | 43.78 ± 6.41 | 17.90 ± 4.92 | 63.19 ± 6.11 | 29.00 ± 2.45 | 47.32 ± 2.79 | 32.07 ± 22.91 |
| PNA | 66.92 ± 4.06 | 20.47 ± 6.80 | 69.29 ± 3.23 | 49.01 ± 4.75 | 50.32 ± 3.14 | 46.19 ± 7.38 |
| GenAgg | 55.68 ± 4.98 | 22.55 ± 9.02 | 50.50 ± 12.18 | 23.01 ± 4.95 | 51.15 ± 10.55 | 35.16 ± 10.81 |
| RGCNE | 76.08 ± 3.55 | **68.90 ± 3.86** | 75.37 ± 2.64 | 55.26 ± 5.91 | 72.43 ± 3.93 | 39.52 ± 1.59 |
| Multi-GIN | 61.02 ± 2.60 | 33.61 ± 3.44 | 69.77 ± 3.89 | 36.43 ± 6.98 | 76.68 ± 4.55 | 47.78 ± 33.73 |
| Multi-PNA | 66.16 ± 6.59 | 43.99 ± 8.72 | 78.32 ± 5.42 | 67.22 ± 3.31 | 74.46 ± 3.07 | **72.68 ± 6.25** |
| Multi-GenAgg | 64.66 ± 5.54 | 49.55 ± 12.38 | 67.45 ± 0.78 | 48.35 ± 1.73 | 76.83 ± 2.88 | 58.54 ± 5.32 |
| Multi-FraudGT | **80.04 ± 1.36** | 68.07 ± 3.34 | 81.18 ± 1.08 | 73.24 ± 1.42 | 80.00 ± 3.49 | 63.38 ± 3.39 |
| MEGA-GIN | 70.11 ± 4.23 | 63.95 ± 4.29 | 74.16 ± 4.91 | 65.61 ± 9.19 | 70.35 ± 6.59 | 38.35 ± 5.36 |
| MEGA-PNA | 76.90 ± 4.05 | 66.26 ± 8.82 | 84.26 ± 0.62 | **75.74 ± 2.75** | **83.81 ± 1.27** | 57.28 ± 8.92 |
| MEGA-GenAgg | 78.27 ± 2.46 | 66.16 ± 1.22 | **85.57 ± 1.29** | 72.09 ± 0.68 | 77.46 ± 3.33 | 60.49 ± 08.77 |
| MEGA-RGCNE | 75.05 ± 3.87 | 58.49 ± 12.59 | 82.28 ± 2.12 | 74.75 ± 2.86 | 67.12 ± 7.39 | 69.26 ± 11.00 |

## E.2 Capacity-matched Comparisons

To complement Table 6 in Section 4.4, we perform *capacity-matched* comparisons to isolate the effect of the proposed neighbor-aware aggregation. Specifically, we scale the baseline models, GIN and PNA, by increasing their parameter counts to closely match those of MEGA-GIN and MEGA-PNA, respectively. * denotes the capacity-matched variant with adjusted hidden dimension. This setup allows us to separate performance gains arising from increased model capacity from those introduced by neighbor-aware aggregation.

The results are presented in Table 18. While increasing the parameter budget yields modest improvements in some cases, most notably on AML Small-HI, where GIN and PNA improves by approximately 5 points in minority-class F1, these gains are inconsistent across datasets and occasionally lead to performance degradation. In contrast, augmenting the same backbones with neighbor-aware aggregation (MEGA-GIN and MEGA-PNA) results in substantial and consistent improvements across all datasets. On average, we observe a relative gain of around 40% in minority-class F1 across both architectures.

These findings indicate that the observed performance improvements cannot be attributed solely to increased model capacity. Instead, they provide strong evidence that the proposed neighbor-aware aggregation is the primary driver of the gains.

Table 17: Recall scores (%) on AML edge classification task. Best result is indicated with **bold**.

| Model | Small HI | Small LI | Medium HI | Medium LI | Large HI | Large LI |
|---|---|---|---|---|---|---|
| GIN | $50.06 \pm 2.62$ | $23.14 \pm 2.70$ | $55.67 \pm 5.77$ | $22.88 \pm 2.72$ | $52.68 \pm 0.32$ | $2.72 \pm 2.02$ |
| PNA | $59.65 \pm 1.82$ | $22.94 \pm 1.77$ | $64.79 \pm 3.07$ | $23.73 \pm 2.58$ | $63.19 \pm 0.74$ | $16.85 \pm 1.93$ |
| GenAgg | $57.52 \pm 2.59$ | $22.37 \pm 3.18$ | $60.06 \pm 3.42$ | $19.92 \pm 4.00$ | $54.86 \pm 2.92$ | $5.37 \pm 1.91$ |
| RGCNE | $55.10 \pm 2.91$ | $25.69 \pm 1.17$ | $58.29 \pm 0.81$ | $26.52 \pm 1.07$ | $48.91 \pm 2.34$ | $16.57 \pm 0.90$ |
| Multi-GIN | $64.49 \pm 2.47$ | $31.37 \pm 2.46$ | $66.03 \pm 2.01$ | $28.47 \pm 3.36$ | $67.06 \pm 1.42$ | $6.83 \pm 7.25$ |
| Multi-PNA | $69.06 \pm 1.51$ | $29.95 \pm 1.86$ | $68.12 \pm 2.33$ | $32.55 \pm 0.67$ | $70.45 \pm 0.64$ | $21.94 \pm 2.11$ |
| Multi-GenAgg | $65.32 \pm 2.71$ | $29.55 \pm 1.95$ | $65.56 \pm 2.84$ | $30.94 \pm 0.41$ | $67.27 \pm 0.79$ | $22.24 \pm 0.91$ |
| Multi-FraudGT | $\mathbf{72.02 \pm 0.96}$ | $34.42 \pm 1.16$ | $71.40 \pm 0.64$ | $32.14 \pm 0.82$ | $68.05 \pm 1.42$ | $24.70 \pm 0.43$ |
| MEGA-GIN | $71.74 \pm 1.64$ | $33.25 \pm 0.87$ | $67.77 \pm 1.40$ | $35.71 \pm 0.60$ | $63.64 \pm 3.15$ | $6.91 \pm 1.13$ |
| MEGA-PNA | $71.48 \pm 1.32$ | $\mathbf{35.89 \pm 0.65}$ | $\mathbf{73.07 \pm 0.39}$ | $\mathbf{36.69 \pm 0.72}$ | $\mathbf{71.15 \pm 0.43}$ | $\mathbf{29.10 \pm 0.74}$ |
| MEGA-GenAgg | $71.14 \pm 1.72$ | $35.62 \pm 0.50$ | $69.53 \pm 1.40$ | $32.60 \pm 0.18$ | $70.16 \pm 1.01$ | $24.71 \pm 1.49$ |
| MEGA-RGCNE | $66.83 \pm 0.96$ | $32.14 \pm 0.97$ | $68.09 \pm 1.37$ | $28.48 \pm 0.76$ | $68.26 \pm 1.42$ | $17.09 \pm 0.76$ |

Table 18: Parameter-matched comparisons of MEGA-GNN with only neighbor-aware aggregation and baselines. K denotes $10^3$.

| Ablation | # of params | AML Small HI | AML Small LI | AML Medium HI | AML Medium LI | ETH |
|---|---|---|---|---|---|---|
| GIN | 69.6K | $46.50 \pm 4.11$ | $19.93 \pm 3.55$ | $58.65 \pm 2.50$ | $25.36 \pm 1.49$ | $42.33 \pm 3.70$ |
| GIN* | 86.23K | $51.70 \pm 4.87$ | $17.72 \pm 7.28$ | $55.38 \pm 5.47$ | $22.62 \pm 3.34$ | $44.46 \pm 1.33$ |
| MEGA-GIN (GIN with neighbor-aware Agg.) | 86.27K | $69.98 \pm 2.02$ | $41.45 \pm 2.13$ | $69.50 \pm 0.85$ | $44.69 \pm 0.13$ | $43.56 \pm 2.67$ |
| PNA | 32.2K | $62.96 \pm 1.43$ | $21.02 \pm 4.05$ | $66.87 \pm 1.87$ | $31.79 \pm 2.30$ | $53.93 \pm 2.45$ |
| PNA* | 40.7K | $67.40 \pm 1.85$ | $26.09 \pm 0.52$ | $68.17 \pm 2.35$ | $32.31 \pm 1.06$ | $53.49 \pm 2.29$ |
| MEGA-PNA (PNA with neighbor-aware Agg.) | 41.8K | $73.65 \pm 0.36$ | $43.77 \pm 1.53$ | $76.77 \pm 0.19$ | $\underline{48.08 \pm 0.32}$ | $59.13 \pm 0.51$ |

### E.3 Comparison with DIAM

Table 19: Comparison with DIAM on AML edge classification task.

| Model | Small HI | Small LI | Medium HI | Medium LI |
|---|---|---|---|---|
| DIAM | $51.82 \pm 6.09$ | $9.80 \pm 1.50$ | $42.37 \pm 2.39$ | $4.58 \pm 0.99$ |
| MEGA-PNA | $74.01 \pm 1.55$ | $46.32 \pm 2.07$ | $78.26 \pm 0.11$ | $49.40 \pm 0.54$ |

On the four AML subsets evaluated, MEGA-PNA clearly outperforms DIAM (see Table 19), demonstrating the advantages of modeling edge-attributed multigraphs to capture complex transaction patterns. Unlike DIAM, which is a specialized solution for ETH node classification task Chen et al. (2021a), our model MEGA-GNN is a general-purpose architecture capable of both node and edge classification on multigraphs. It generates expressive edge embeddings and propagates them through the message passing process.

### E.4 Wilcoxon Signed-Rank Tests

To assess the statistical significance of the AML results reported in Section 4.3.1, we conduct two-sided Wilcoxon signed-rank tests over the six per-dataset paired differences for each model comparison. At $n=6$, the minimum achievable $p$-value is 0.031, corresponding to $W=0$ (all six differences in the same direction). Table 20 summarises the test statistics and $p$-values.

MEGA-GNN variants attain the minimum achievable $p$-value ($W=0$, $p=0.031$) against all four corresponding base models, confirming consistent improvement across every AML dataset. Against Multi-GNN variants, MEGA-PNA and MEGA-GenAgg also reach $p=0.031$, while MEGA-GIN yields $p=0.062$, reflecting one dataset where Multi-GIN is marginally stronger. When MEGA-PNA is compared against the strongest

Table 20: Two-sided Wilcoxon signed-rank test results for MEGA-GNN comparisons on six AML datasets ($n$=6). $W$ = min(sum of positive ranks, sum of negative ranks); lower W indicates more consistent improvement.

| Comparison | $W$ | $p$-value |
|---|---|---|
| MEGA-GIN vs. GIN | 0 | 0.031 |
| MEGA-PNA vs. PNA | 0 | 0.031 |
| MEGA-GenAgg vs. GenAgg | 0 | 0.031 |
| MEGA-R-GCNE vs. R-GCNE | 0 | 0.031 |
| MEGA-GIN vs. Multi-GIN | 1 | 0.062 |
| MEGA-PNA vs. Multi-PNA | 0 | 0.031 |
| MEGA-GenAgg vs. Multi-GenAgg | 0 | 0.031 |
| MEGA-PNA vs. best non-MEGA baseline | 2 | 0.093 |

non-MEGA baseline on each dataset, the test gives $W$=2, $p$=0.093, with the 5-of-6 sign pattern providing the primary evidence of consistent improvement.

### E.5 Why Is MEGA-GNN's Gain Larger on LI Datasets?

Figure 10 reports the relative F1 improvement of MEGA variants across the four backbones (GIN, PNA, GenAgg, R-GCNE), grouped by illicit ratio (LI vs. HI) and scale. The left panel compares each MEGA-GNN to its base GNN; the right to the corresponding Multi-GNN. Bars show the mean across backbones; error bars the standard deviation; points the per-backbone values.

The mean relative gain is larger on the LI datasets than on the corresponding HI datasets at every scale and against both reference points. Per-backbone values vary in magnitude, and not every backbone preserves the LI–HI ordering individually, but the ordering holds on average. This is consistent with the higher fraction of illicit edges placed on node pairs with multi-edges in the LI datasets (Table 13).

Figure 10: Relative F1 improvement of MEGA-GNNs on the AML datasets by illicit ratio (LI, blue; HI, red) and scale. *Left:* vs. base GNN. *Right:* vs. Multi-GNN. Bars: mean across four backbones (GIN, PNA, GenAgg, R-GCNE). Error bars: standard deviation. Points: per-backbone values.

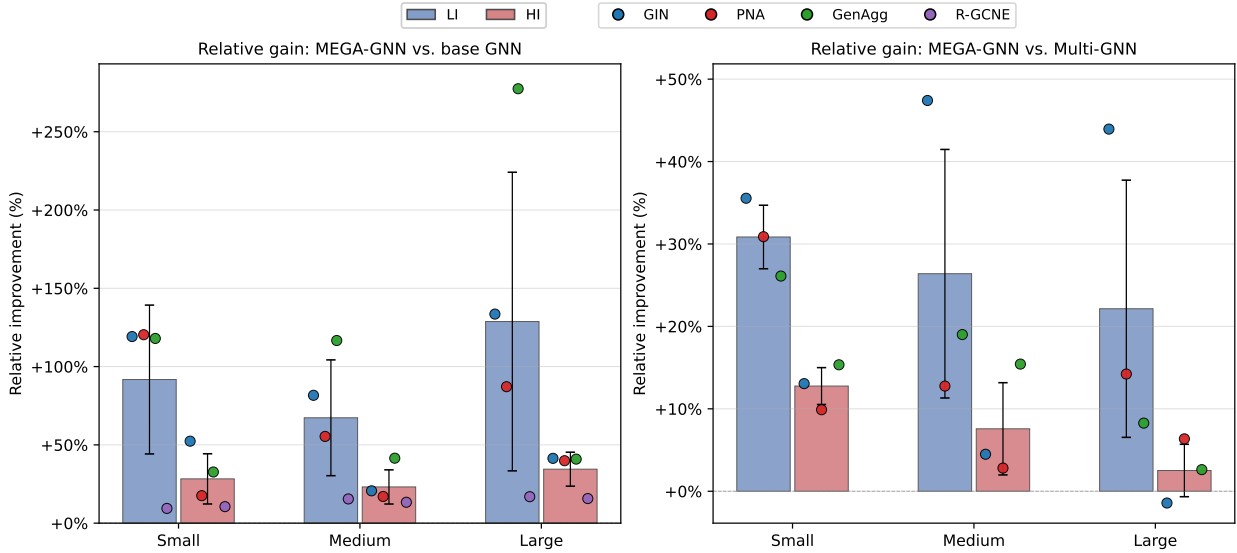

## F   Broader Impact

This work contributes effective graph machine learning techniques for financial crime analysis by addressing the specific challenges posed by multigraph structures in financial transaction networks. Our model learns to detect illicit behavior directly from data, rather than relying on predefined, rule-based systems. This end-to-end approach improves adaptability and detection performance.

By enabling more accurate detection of illicit activity, our model has the potential to support financial institutions and regulatory bodies in identifying and preventing illicit financial behavior, such as money laundering and fraud. This may lead to stronger financial oversight, reduced criminal financing, and overall societal benefit through enhanced economic transparency and security.

At the same time, deploying detection models in financial systems carries real risks. False positives can lead to frozen accounts, denied transactions, or reputational harm for the individuals and organizations involved. Biases in historical transaction data, shaped by past reporting and investigative practices, may also be inherited or amplified by a learned model. This can result in disproportionate scrutiny of particular populations or business segments. A further limitation is interpretability: like most graph neural networks, our model is not inherently explainable, which can be problematic in regulated settings. For these reasons, we view our method as a tool to *support* expert human judgment rather than replace it. Practitioners should pair it with fairness auditing, post-hoc interpretability techniques, and human-in-the-loop review before acting on its outputs.

## G   LLM Usage

LLMs were used solely as a writing assistant to polish the language of this manuscript, such as checking grammar and improving clarity of expression. They were not used extensively.

