# OpenReview forum: "MEGA: Message Passing Neural Networks for Multigraphs with EdGe Attributes"
_TMLR — Decision pending for TMLR_

### Review · Reviewer_LxHX · 2026-04-21

**Summary Of Contributions:**

The paper introduces a modification to message passing in graph neural networks tailored to edge-attributed multigraphs, where multiple edges can connect the same pair of nodes. The key idea is to change how aggregation is performed: instead of pooling all incoming edges in a single step, the method first aggregates edges separately for each neighbor and then aggregates across neighbors. This “neighbor-aware” aggregation is presented as a simple but general operator that can be inserted into existing GNN architectures. The authors provide a theoretical argument that this two-stage aggregation is more expressive than standard single-stage aggregation for a class of aggregators, and they instantiate it in a framework called MEGA-GNN. Empirically, they evaluate the approach on synthetic data designed to test per-neighbor statistics and on several real-world datasets from social interaction and financial transaction domains, showing consistent improvements over baseline GNN variants.

Strengths: The paper is built around a clear and well-motivated idea: separating edge-level and neighbor-level aggregation to better handle multigraph structure. This is conceptually simple, easy to integrate, and addresses a limitation of standard message passing. The theoretical analysis explains why the proposed operator is more expressive (improves the conceptual contribution). The empirical evaluation covers both controlled synthetic settings, multiple real-world datasets, and includes comparisons against some baselines. The method is also presented as backbone-agnostic, which increases its potential applicability.

Weaknesses: The main weakness is a misalignment between the strength of the claims and the strength of the evidence. The paper suggests fairly general advantages, but the experimental evidence does not fully support such broad conclusions. In particular, the synthetic benchmark is closely aligned with the proposed mechanism, which limits its value as independent validation. The empirical section lacks proper across-dataset statistical analysis, making it difficult to assess whether improvements are consistent or dataset-specific. The ablation studies do not fully isolate the mechanism, so it remains unclear how much of the gain is due specifically to the proposed aggregation versus other factors such as increased model capacity. Finally, the paper does not provide a clear analysis of when the method is expected to help, leaving the reader without a concrete understanding of the conditions under which the approach should be preferred.

**Audience:**

Yes

**Audience Explanation:**

Yes, some people in the TMLR audience would likely be interested in this paper.

The topic is relevant to researchers working on graph neural networks, especially those dealing with more complex graph structures like multigraphs with edge attributes. The paper proposes a simple modification to message passing that can be added to existing models, which makes it practical and easy to try. The idea of separating aggregation over edges and over neighbors is also intuitive and could influence how people think about designing GNN architectures.

In addition, the paper connects a clear modeling issue with a concrete solution and shows empirical improvements on real datasets, including applications like financial transactions and social networks. This makes it interesting not only from a theoretical point of view but also for applied researchers.

**Broader Impact Concerns:**

The paper focuses on improving graph neural networks for structured data and does not introduce any direct or obvious ethical risks at the method level. However, the application domains used in the experiments, such as financial transaction networks and fraud detection, can have real-world implications. Models in these settings may affect decisions about individuals or organizations, including flagging accounts as suspicious, which raises concerns about false positives, fairness, and potential bias in the data.

The paper does not appear to discuss these aspects in detail. In particular, there is no analysis of how errors might impact users, whether the method could amplify existing biases in transaction data, or how it should be used responsibly in high-stakes environments. There is also no discussion of transparency or interpretability, which could be important in contexts where decisions need to be explained.

**Claims And Evidence:**

No

**Claims Explanation:**

The claims are only partially supported by the evidence.

The paper does a good job showing that the method improves performance on the datasets it tests. The experiments are clearly presented, repeated multiple times, and the results are consistent. The theoretical argument also supports the idea that the proposed aggregation can represent more complex patterns than standard approaches, at least under certain assumptions. So for a narrow claim that the method can work better in the tested settings, the evidence is reasonable and credible.

However, the paper suggests stronger conclusions than the evidence really supports. The synthetic experiment is designed around the exact type of structure the method is meant to capture, so it mainly shows that the method works in a situation built to match its design. This makes it less convincing as general evidence. The results on real datasets are promising, but there is no proper statistical comparison across datasets, so it is unclear how stable or consistent the improvements are. The ablation studies show that the components matter, but they do not fully prove that the specific idea proposed by the authors is the true reason for the gains. Finally, the paper does not explain clearly when the method should be expected to help, which makes the conclusions harder to interpret.

Overall, the evidence is clear and correct, and it supports a limited claim about improved performance in the tested scenarios. But it is not strong enough to justify broader claims about general advantage or to fully validate the proposed explanation for why the method works.

**Requested Changes:**

1. The central claim needs to be made more precise and aligned with the actual evidence provided. At the moment, the paper suggests a fairly broad conclusion about general superiority, but the experiments mainly support improvements under specific conditions. The authors should explicitly separate what is being claimed about predictive performance, what is being claimed about the underlying mechanism, and under which conditions these claims are expected to hold. Narrowing the claim to match the evidence would make the contribution more defensible and prevent overinterpretation.
   Specific experiment: include a controlled comparison where datasets are grouped by a key property (for example, low vs high edge multiplicity) and report performance separately for each group, showing explicitly that gains are concentrated in certain regimes rather than uniform.

2. The experimental results need proper across-dataset statistical analysis to justify any claim of general improvement. Right now, results are presented as averages or tables, but there is no formal assessment of whether the observed gains are consistent across datasets or just driven by a few cases. The authors should report per-dataset differences, show how often their method wins or loses, and include a simple statistical comparison across datasets. Without this, the paper cannot support conclusions beyond local benchmark improvements.
   Specific experiment: for each dataset, compute the difference in performance between MEGA and the strongest baseline, then report win/loss counts, average rank across datasets, and include a simple paired statistical test (for example, a sign test or Wilcoxon signed-rank test) to assess whether improvements are consistent.

3. The synthetic benchmark should not be treated as primary evidence for the method’s effectiveness. As designed, it directly encodes the type of structure the method is built to capture, which risks circular validation. The authors should clearly position this experiment as illustrating the mechanism rather than proving general usefulness, and complement it with evidence showing that similar effects appear in real data where the structure is not artificially constructed.
   Specific experiment: construct a modified real-data experiment where edge multiplicity is artificially perturbed (for example, by randomly collapsing multiple edges into a single edge or duplicating edges), and show how performance changes. This would test whether the advantage correlates with the presence of multi-edge structure in realistic settings.

4. The mechanism claim requires stronger validation through more careful ablations. The current ablations show that components matter, but they do not fully isolate whether the proposed aggregation strategy itself is responsible for the gains. The authors should include comparisons against alternative designs and ensure that differences are not simply due to increased capacity or architectural changes. This would provide clearer evidence that the proposed idea, rather than incidental factors, is driving the improvements.
   Specific experiment: introduce a baseline that uses a two-stage aggregation with random grouping of edges (instead of grouping by neighbor), or a capacity-matched model with the same number of parameters but standard aggregation, and compare performance. If the proposed method still outperforms these controls, it would better isolate the effect of the neighbor-aware design.

5. The paper needs to explain when and why the method works, not just show that it works. At present, the results demonstrate improvements but do not analyse the conditions under which these improvements appear or disappear. The authors should relate performance to properties of the graphs, such as the number of repeated edges or the variability of edge attributes, and identify regimes where the method is most beneficial. This would turn the empirical section from a collection of results into a coherent explanation of the method’s behaviour.
   Specific experiment: for each dataset, compute summary statistics such as average edge multiplicity, variance of edge attributes per node, or degree distribution, and plot performance improvement as a function of these quantities. This would reveal whether gains increase with stronger multi-edge structure and provide a concrete explanation of when the method is most useful.

---

> ### Author Response · Authors · 2026-05-21
> **Official Response by Authors (1/3)**
>
> We thank the reviewer for the thoughtful and constructive feedback. The comments helped us sharpen the scope of our claims, strengthen the empirical evidence, and better isolate the contribution of our proposed mechanism. We address each point in detail below. All changes in the revised manuscript are marked in **blue** for ease of review.
>
> ---
> ### (1) Sharpening the central claim
> On the scope of the central claim. We thank the reviewer for the careful read. Throughout the manuscript we have tried to tie each claim to the specific benchmark, task, and baselines it refers to, and to avoid extrapolating beyond what the experiments support. On re-reading the manuscript with the reviewer's concern in mind, we identified a few places where the phrasing could be sharpened, and we revised them accordingly:
> - Section 4.1 (synthetic diagnostic). We replaced "clearly outperforms all baselines" with "achieves the lowest MAE".
> - Introduction. We removed the phrasing "achieves strong performance across a diverse set of tasks" and "consistently improves GNN performance". The empirical claim now reads as an evaluation on the three listed benchmarks, with improvements stated relative to the corresponding baselines.
> - Conclusion. We replaced "achieves substantial gains in minority-class F1 over baseline GNNs, Multi-GNN, and Multi-FraudGT" with the precise paired-comparison summary: improvement over baseline GNNs across all evaluated dataset variants, and over Multi-GNN in 17 of 18 paired comparisons. We also specify the regime where the gains are largest (the low-illicit AML datasets, where a larger fraction of illicit edges sits on node pairs with multi-edges, see Appendix C, Table 13).
>
> We hope these revisions resolve the reviewer's concern. If there are other phrases where the reviewer feels the claim is broader than the evidence, we would be grateful if they could point us to them and we will be happy to revise those as well.
>
> On the controlled grouping experiment. We appreciate the suggestion to relate MEGA's gains to structural properties of the datasets; we discuss this analysis in detail in our response to Comment 5. Grouping the AML datasets by a single graph statistic, such as mean edge multiplicity, is not truly controlled, since other statistics co-vary with it. As mean edge multiplicity increases across datasets, so do its standard deviation and other related structural quantities, so any single-variable grouping conflates several factors at once.
>
> The LI/HI pairs offer a more direct comparison: within each pair, the graphs are closely matched in degree distribution and multi-edge multiplicity, and differ mainly in how illicit edges are placed across multi-edge pairs (see Table 13 in Appendix C). Using this grouping, we find that MEGA's gains are, on average, larger in the LI regime than in the corresponding HI regime across all three scales (see Appendix E.5). We have updated the manuscript to present this pattern explicitly as an empirical observation rather than a causal claim.
>
> ---
> ### (2) Across-dataset statistical analysis
>
> The requested analyses have been added to Section 4.3.1, Table 4, Table 15 (Appendix E) and Appendix E.4.
>
> **Win/loss counts and average rank.** Across the six AML datasets, MEGA outperforms its corresponding base GNN in **24 of 24** head-to-head comparisons (4 baselines × 6 datasets), and outperforms Multi-GNN in **17 of 18**. Against the strongest non-MEGA baseline on each dataset, MEGA-PNA wins on **5 of 6**; the single loss is on AML-Small-HI, where Multi-FraudGT is narrowly better. These counts were already discussed in the original AML results paragraph; the revision additionally reports an average rank across the six AML datasets, with MEGA-PNA achieving the best rank of 1.33, followed by MEGA-GenAgg (2.33) and Multi-FraudGT (2.83). MEGA variants occupy four of the top six positions overall.
>
> **Wilcoxon signed-rank test.** Following standard practice for comparing classifiers across multiple datasets [1], we apply the two-sided Wilcoxon signed-rank test to the six AML datasets, which share a matched protocol (edge classification, minority-class F1, identical temporal splits). The other benchmarks differ in task or metric (JODIE uses ROC-AUC for user state-change prediction, and ETH is a single node-classification dataset) so we report them separately (Tables 2 and 3). Per-pair two-sided p-values are reported in Appendix E.4, and discussed in Section 4.3.1.
>
> [1] Demšar, Janez. "Statistical comparisons of classifiers over multiple data sets." Journal of Machine learning research 7.Jan (2006).

---

> > ### Author Response · Authors · 2026-05-21
> > **Official Response by Authors (2/3)**
> >
> > ### (3) Synthetic benchmark and multi-edge perturbation on real data
> >
> > We have revised Section 4.1 accordingly, the section is now titled "Controlled Diagnostic of Per-Neighbor Aggregation" and explicitly frames the experiment as illustrating  the mechanism.
> >
> > To address the reviewer's concrete suggestion, we ran the proposed multi-edge perturbation experiment on the ETH dataset. At collapse rates of 0%, 25%, 50%, 75%, and 100%, we randomly merge all transactions between a fraction of node pairs into a single edge (using the mean amount and earliest timestamp), leaving all other graph properties unchanged. We compare PNA against MEGA-PNA with neighbor-aware aggregation only (no bi-directional MP), so that multi-edge  presence is the only variable.
> >
> > We focus on the ETH dataset because its labels are defined at the node level. On the AML datasets, labels are edge-level, so collapsing multi-edges would require a label-merging rule (e.g., "illicit if any merged edge is illicit"), which would itself alter the per-class label distribution and confound the comparison. Keeping labels invariant to the perturbation is essential for the experiment to cleanly isolate the effect of multi-edge structure.
> >
> > **Result.** PNA's F1 is essentially flat across all collapse rates (~53%), consistent with single-stage aggregation being unable to exploit multi-edge structure even where it exists. MEGA-PNA, in contrast, degrades from 59.13% at 0% collapse to 52.67% at full collapse, and its advantage over PNA shrinks from +5.20 pp to -0.44 pp:
> >
> > | Collapse rate | PNA          | MEGA-PNA     | Gap (pp) |
> > |---------------|--------------|--------------|----------|
> > | 0%            | 53.93 ± 2.45 | 59.13 ± 0.51 | +5.20    |
> > | 25%           | 53.95 ± 3.48 | 56.39 ± 1.29 | +2.44    |
> > | 50%           | 52.94 ± 3.71 | 56.37 ± 2.42 | +3.43    |
> > | 75%           | 53.92 ± 4.68 | 53.50 ± 0.90 | -0.42    |
> > | 100%          | 53.11 ± 1.78 | 52.67 ± 1.66 | -0.44    |
> >
> > The gap closes precisely when multi-edge structure is removed, providing direct real-data evidence that MEGA-PNA's improvement is driven by its ability to exploit multi-edges, rather than by capacity or other architectural factors. Together with the capacity-matched results discussed in our response to Comment 4, this isolates the proposed mechanism as the source of the gains.
> >
> > We have added this experiment as Table 5 and a corresponding discussion in Section 4.4.
> >
> > ---
> > ### (4) Isolating the mechanism: capacity-matched baselines
> > To isolate the effect of the aggregation mechanism from raw model capacity, we added the capacity-matched baselines suggested by the reviewer: standard GIN and PNA scaled in hidden dimension to approximately match the parameter counts of MEGA-GIN and MEGA-PNA. The new results are reported in Table 18 (Appendix E.2) and summarized in Section 4.4. For transparency, we also added the parameter counts of all ablated configurations to Table 6.
> >
> > **Result.** Capacity matching yields only small and inconsistent improvements over the base GIN/PNA models, a few F1 points in either direction. In contrast, replacing the single-stage aggregation with our neighbor-aware aggregation yields gains of 10–20 F1 points. The improvement from the proposed aggregation is thus substantially larger than what capacity alone can explain.
> >
> > On the random-grouping alternative. We chose the capacity-matched control because it directly targets the capacity-vs-mechanism question raised in the review. If the reviewer would also find a random-grouping comparison informative, we are glad to add it during the discussion period.

---

> > > ### Author Response · Authors · 2026-05-21
> > > **Official Response by Authors (3/3)**
> > >
> > > ### (5) When and why the method works
> > > Following this feedback, we extend our empirical analysis to explicitly relate performance gains to structural properties of the AML datasets.
> > >
> > > We compute summary statistics for each dataset (Table 13 in Appendix C), including the fraction of edges participating in multi-edge interactions and the average edge multiplicity. Across all datasets, we observe a pronounced multigraph regime: over 90% of edges participate in repeated interactions, with average multiplicities ranging from approximately 5 to over 21.
> > >
> > > Relating performance gains to any single summary statistic in isolation is challenging, since these graph properties co-vary across datasets. For example, as mean edge multiplicity increases, so do its standard deviation and other related quantities. Attributing performance improvements to a single factor would therefore conflate multiple structural effects. The LI/HI dataset separation nevertheless reveals two empirical trends. First, when fixing the scale and varying the illicit ratio, MEGA variants yield, on average, larger F1 improvements on the LI variants than on the corresponding HI variants, both relative to the base GNNs and to the Multi-GNN baselines. Figure 10 (Appendix E.5) summarizes this effect across the four backbones (GIN, PNA, GenAgg, and R-GCNE). The larger gains on LI datasets is consistent with a structural difference visible in Table 13(in Appendix C): across all three scales, the fraction of illicit edges placed on node pairs connected by multiple transactions higher in the LI datasets (0.38, 0.44, 0.70) than in the corresponding HI datasets (0.27, 0.31, 0.46), giving MEGA's neighbor-aware aggregation more multi-edge structure to exploit on the LI side.
> > >
> > > Second, when fixing the illicit ratio and varying the scale, the F1 gain of MEGA-PNA over the strongest non-MEGA baseline increases monotonically with scale on the HI variants (−1.8 → +2.1 → +3.9 F1 points). The LI variants exhibit a non-monotonic trend, so we refrain from making a broader claim about scale alone. We present these observations as empirical patterns rather than causal conclusions tied to a single graph statistic.
> > >
> > > This structural analysis complements the mechanism-isolation experiment reported in our response to Comment 3 (the multi-edge collapse experiment on ETH), which directly perturbs multi-edge structure on real data and shows that MEGA-PNA's advantage shrinks as multi-edges are removed.
> > >
> > > ---
> > > ### Broader Impact
> > >
> > > We have expanded the Broader Impact (Appendix F) section with an additional paragraph that explicitly discusses (i) the real-world consequences of false positives on individuals and organizations, (ii) the risk that biases in historical transaction data may be inherited or amplified by the model, (iii) the limited interpretability of GNNs in regulated settings, and (iv) our recommendation that the method be used as decision support with human-in-the-loop review rather than as an autonomous decision-maker. We hope this addresses the reviewer's concerns.

---

> > > > ### Comment · Reviewer_LxHX · 2026-05-21
> > > >
> > > > Thank you,  I appreciate that the authors have taken the concerns seriously and have made substantial changes to better align the claims with the evidence. In particular, I find the added discussion of the scope of the claims, the across-dataset statistical analysis, the structural analysis of the AML datasets, and the capacity-matched comparisons helpful. These additions make the empirical argument considerably more convincing and clarify when the proposed neighbor-aware aggregation is expected to be beneficial.
> > > >
> > > > The remaining point I would encourage the authors to address, if feasible, is the additional control mentioned in your response: a random-grouping or otherwise non-neighbor-based two-stage aggregation baseline. The current capacity-matched experiment helps rule out increased parameter count as the main explanation, but a random-grouping control would more directly test whether the specific neighbor-based grouping is essential, rather than the gain coming from a generic two-stage aggregation mechanism.
> > > >
> > > > I do not see this as a major remaining weakness, but including this control, even on a subset of representative datasets, would strengthen the causal interpretation of the results and make the mechanism-isolation argument more complete. Overall, I am satisfied that the authors have responded well to my main concerns.

---

> > > > > ### Author Response · Authors · 2026-06-03
> > > > > **Official Response by Authors**
> > > > >
> > > > > We thank the reviewer for the positive assessment. To address the remaining point, we have added the random-grouping control.
> > > > >
> > > > > **Random-grouping control.** To test whether the specific `(src, dst)` grouping in our neighbor-aware aggregation drives the observed gains, rather than the two-stage mechanism itself, we introduce a random-grouping control. For each destination node, we randomly permute the `src` column of its incoming edges; edge features (amount, timestamp, etc.) remain pinned to their original edge positions. The permutation is constrained to preserve, for each destination, the multiset of (source, count) pairs: if M0 originally receives 2 edges from A0 and 1 from A1, the permuted version still has groups of size (2, 1), but the sources are reassigned to different edges. The model therefore sees the same neighborhood, the same group-size distribution, and the same transaction-level features, it simply can no longer attribute which sender produced which transaction, nor which transactions are grouped together.
> > > > >
> > > > > Concretely, for destination M0:
> > > > >
> > > > > ```
> > > > > BEFORE shuffle (dst = M0):
> > > > >   edge pos   src    amount
> > > > >      0        A0     $300    ┐ group (A0, M0)
> > > > >      1        A0     $120    ┘
> > > > >      2        A1      $45    ── group (A1, M0)
> > > > >
> > > > > AFTER shuffle (dst = M0):
> > > > >   edge pos   src    amount
> > > > >      0        A1     $300    ← $300 was A0's, now attributed to A1
> > > > >      1        A0     $120
> > > > >      2        A0      $45    ← $45 was A1's, now attributed to A0
> > > > > ```
> > > > > The group sizes (2, 1) are preserved, but the assignment of edges to groups is randomized. Because the architecture, parameter count, and message contents are identical to our proposed model, a performance drop under this control isolates the contribution of correct neighbor-based grouping from the generic two-stage aggregation mechanism.
> > > > >
> > > > > **Results.** We evaluate the control on Small-HI for GIN and PNA backbones. Replacing neighbor-based grouping with the random-grouping control degrades performance in both cases, confirming that neighbor-aware aggregation is responsible for the observed gains.
> > > > >
> > > > > |Model|Grouping|F1 (Small-HI)|
> > > > > |:--|:--|:-:|
> > > > > |MEGA-GIN|neighbor-based (ours)|**70.83 ± 2.19**|
> > > > > |MEGA-GIN|random (control)|61.21 ± 3.21|
> > > > > |MEGA-PNA|neighbor-based (ours)|**74.01 ± 1.55**|
> > > > > |MEGA-PNA|random (control)|66.89 ± 1.63|
> > > > >
> > > > > The random-grouping control drops F1 by 9.62 points for MEGA-GIN and 7.12 points for MEGA-PNA, with the control's standard deviation also widening for GIN (3.21 vs. 2.19). Both drops are large relative to the standard deviations across seeds, indicating that the gap is not attributable to seed variance.
> > > > >
> > > > > > If the reviewers consider these results important for the completeness of the evaluation, we can conduct the same experiments across all AML Small and AML Medium benchmark datasets and include the results in the revised version of the paper.

---

> > > > > > ### Author Response · Authors · 2026-06-16
> > > > > > **Follow-up: random-grouping control extended to other datasets**
> > > > > >
> > > > > > Following up on our previous response: these experiments have now finished, so we can report the random-grouping control across four AML Small and Medium benchmarks (HI and LI variants) for both the GIN and PNA backbones, rather than the Small-HI subset shown earlier.
> > > > > >
> > > > > > | Model    | Grouping              |     Small-HI     |     Small-LI     |    Medium-HI     |    Medium-LI     |
> > > > > > | :------- | :-------------------- | :--------------: | :--------------: | :--------------: | :--------------: |
> > > > > > | MEGA-GIN | neighbor-based (ours) | **70.83 ± 2.19** | **43.67 ± 0.55** | **70.77 ± 2.76** | **46.05 ± 1.64** |
> > > > > > | MEGA-GIN | random (control)      |   61.21 ± 3.21   |   27.95 ± 2.33   |   67.26 ± 1.08   |   9.66 ± 9.79    |
> > > > > > | MEGA-PNA | neighbor-based (ours) | **74.01 ± 1.55** | **46.32 ± 2.07** | **78.26 ± 0.11** | **49.40 ± 0.54** |
> > > > > > | MEGA-PNA | random (control)      |   66.89 ± 1.63   |   36.62 ± 2.46   |   69.14 ± 0.41   |   39.73 ± 1.15   |
> > > > > >
> > > > > > The random-grouping control degrades performance in all eight settings, indicating that the gains stem from correct neighbor-based grouping rather than from the generic two-stage mechanism. The effect is consistent across backbones and datasets and is most pronounced on the LI variants. On Medium-LI, MEGA-GIN's F1 drops from 46.05 to 9.66 under the control, with a standard deviation (±9.79) that exceeds its mean; the control's results vary widely across seeds, whereas our neighbor-based model stays stable (±1.64). We will include the full table in the revised version.

---

### Review · Reviewer_pyax · 2026-05-10

**Summary Of Contributions:**

This paper tackles an important and underexplored setting: edge-attributed multigraphs where repeated interactions between the same node pair matter. The proposed two-stage "neighbor-aware" aggregation is intuitive, the large-scale AML/ETH evaluations are practically relevant, and the component ablations are directionally encouraging.

**Audience:**

Yes

**Audience Explanation:**

Multi-relational GNN is an important research topic, and repeated interactions between the same node pair are genuinely important in finance and social interaction graphs.

**Claims And Evidence:**

Yes

**Claims Explanation:**

This paper firstly identifies the port-number (some sort of auxiliary identifier) of Multi-GNN that breaks permutation equivariance. The two-stage aggregation idea is then proposed to address this issue. The AML experiments are large scale and practically meaningful. The ablations are useful and do show that neighbor-aware aggregation carries much of the gain.

There is one minor point to note, Table 4 evaluates sensitivity to altered port assignments at test time, which is a robustness test, not a direct equivariance test.

**Requested Changes:**

Some generalization claims are broader than the direct evidence. For instance, the social-network evaluation only uses MEGA-PNA vs PNA, Multi-PNA, and JODIE. The paper still describes the method as broadly backbone-agnostic across tasks. That is plausible, but the cross-backbone evidence is much stronger on AML than on the JODIE tasks.

The "state-of-the-art" empirical positioning suffers from a similar issue, the DIAM-on-AML table reports only two AML subsets, but the prose generalizes this to "the illicit transaction detection task".

This is not fatal, but the framing should be a bit more careful.

---

> ### Author Response · Authors · 2026-05-21
> **Official Response by Authors**
>
> We thank the reviewer for the positive assessment and for the careful observations. We address each point in detail below, and all changes in the revised manuscript are marked in **blue** for ease of review.
>
> ---
> ### (1) Scope of Table 4
>
> We thank the reviewer for this precise observation. The reviewer is correct that Table 4 evaluates the sensitivity of the models to port permutations rather than permutation equivariance itself. Equivariance is a property of the function and is established analytically: Proposition 1 shows that MEGA-GNN is permutation equivariant, while Proposition 2 shows that methods relying on multigraph port numbers (Multi-GNN, Multi-FraudGT) are not. We have revised the paragraph heading (now "Robustness to port permutations") and the surrounding text to make this distinction explicit.
>
> ---
> ### (2) Scope of the "backbone-agnostic" claim
>
> To clarify our framing: by “backbone-agnostic” we mean that MEGA-GNN can be applied on top of different GNN backbones to extend them to edge-attributed multigraphs, as stated in the last paragraph of the Introduction. We do not claim that pairing MEGA with every backbone universally yields the best performance across all tasks.
>
> Our claims are tied to the tasks and experimental settings we evaluate. For example in AML benckmark we evaluated four different backbones across six datasets. In all cases, the corresponding MEGA variants outperformed their base GNN backbones. Accordingly, our statement that “these consistent gains across different backbones suggest that improvements come from MEGA-GNN’s neighbor-aware aggregation rather than architecture-specific effects” was intended specifically in the context of the AML experiments, not as a broad claim across all tasks.
>
> To strengthen the evidence on the JODIE benchmarks, we have added GIN, Multi-GIN, and MEGA-GIN results to Table 2. Across all six (backbone × dataset) combinations, the MEGA variants improve over their corresponding base GNN backbones. Furthermore, the best ROC-AUC on each dataset is still achieved by a MEGA variant: MEGA-PNA on MOOC and Wikipedia, and MEGA-GIN on Reddit. We have updated the Setup and Results paragraphs in Section 4.2 accordingly to discuss these new results.
>
> We thank the reviewer for highlighting this point, as it gave us the opportunity to clarify the scope of our claims and strengthen the empirical support in the JODIE evaluation.
>
> ---
> ### (3) DIAM comparison on AML
> We have extended the DIAM comparison with new experiments on the Medium HI and Medium LI datasets, bringing the total to four of six AML subsets. MEGA-PNA outperforms DIAM on all four. We have also revised the discussion in Appendix E.3 to explicitly scope the claim: "On the four AML subsets evaluated, MEGA-PNA clearly outperforms DIAM."

---

### Review · Reviewer_1SzU · 2026-05-18

**Summary Of Contributions:**

This paper introduces MEGA-GNN, a flexible neural network framework designed for graphs with multiple, attribute-rich connections between the same nodes (edge-attributed multigraphs). By utilizing a novel neighbor-aware aggregation method—which processes multi-edge data per neighbor before combining it globally—the framework captures complex interaction statistics that traditional GNNs miss. MEGA-GNN improves learning expressiveness, guarantees structural symmetry (permutation equivariance), and maintains low computational complexity, leading to state-of-the-art performance on financial fraud and social network benchmarks. The main contributions of this paper are the following:

1. Methodological & Architectural Innovation

Neighbor-Aware Aggregation Operator: The authors introduce a novel two-stage operator that first combines multi-edge features for each individual neighbor before aggregating across different neighbors. This distinguishes contributions from different neighbors while preserving dense information from repeated interactions.

This Neighbor-Aware Aggregation Operator is then used to build the so-called MEGA-GNN architecture: a model- and backbone-agnostic message-passing framework specifically built for edge-attributed multigraphs.

2. Theoretical Advancements

Enhanced Expressiveness: The paper proves that the neighbor-aware aggregation design is strictly more expressive than standard single-stage aggregation in edge-attributed multigraph settings.

Permutation Equivariance: The framework preserves permutation equivariance natively without needing pre-computed multigraph port identifiers (which can break equivariance or add overhead).

Computational Efficiency: The authors demonstrate that MEGA-GNN maintains the same asymptotic complexity as standard GNNs with edge updates.

3. Empirical Results

Financial Crime Benchmarks: MEGA-GNN achieved substantial gains in minority-class F1-scores over strong baselines (including standard GNNs, Multi-GNN, and Multi-FraudGT), showing particularly high resilience and advantages under port permutation.

Temporal User-Item Interaction: An adaptation of the framework (MEGA-PNA) consistently improved ROC-AUC performance on temporal datasets, validating the model's effectiveness across diverse real-world domains like social networks and financial transactions.

**Audience:**

Yes

**Audience Explanation:**

Yes. The paper’s contribution of a scalable, permutation-equivariant GNN framework that improves the fundamental expressiveness limitations on multigraphs directly aligns with TMLR’s core focus on representation learning and geometric deep learning.

**Broader Impact Concerns:**

-

**Claims And Evidence:**

Yes

**Claims Explanation:**

- The authors do an excellent job of contextualizing their contributions within the existing literature.

- The methodology is clearly explained and well-supported by intuitive visualizations that make the approach easy to follow.

- Theoretical analysis demonstrates that the proposed method is more expressive than existing models while maintaining a comparable computational complexity.

- The authors conduct extensive experiments on both simulated and real-world datasets. Notably, the evaluation spans a diverse set of tasks and datasets, including both node-level and edge-level prediction targets.

**Requested Changes:**

I thank the authors for this interesting work. Here are some comments/questions on the submission:

- While the text states that MEGA-GNN maintains the same asymptotic complexity as standard GNNs with edge updates, it would be beneficial to include a brief, explicit breakdown of the runtime and memory overhead.

- It would have been very interesting to discuss what types of per-neighbor statistics the traditional single-stage aggregation fails to capture, making the theoretical "strict expressiveness" claim more concrete.

- Related to the previous question, I think a section in the main paper discusing the limitations of the existing approach should be added. (Potentially highlighting some limitation regarding the possibility to parallelized computations compared to other models, or on the set of edge/node statistics that cannot be captured properly with the proposed architecture).

- Currently, the multi-edge aggregation step does not seem to explicitly account for different types of edges. I am curious if the authors have explored or tested a framework where learnable edge-type embeddings are added to the edge features during the multi-edge aggregation phase. Introducing this mechanism could make the model inherently aware of edge types, allowing it to treat different edge features conditionally based on their category.

---

> ### Author Response · Authors · 2026-05-21
> **Official Response by Authors**
>
> We sincerely thank the reviewer for the careful reading of our paper and the positive assessment of our work. We address each comment in turn below, and all changes in the revised manuscript are marked in **blue** for ease of review.
>
> ---
> ### (1) Explicit runtime and memory overhead breakdown
>
> Table 7 reports per-dataset throughput for the ablated models (PNA, MEGA-PNA, MEGA-PNA with bi-directional MP, and MEGA-PNA with ego IDs and bi-directional MP). The paragraph following Table 7 in Section 4.5 explicitly quantifies the runtime overhead: "Neighbor-aware aggregation achieves an average relative improvement of ~40\% in minority-class F1 while incurring only a ~16\% slowdown in inference throughput."
>
> Peak GPU memory consumption. Figure 4 reports peak GPU memory consumption as a function of neighborhood size on AML-Small-HI and AML-Medium-HI. The MEGA-PNA and PNA curves overlap closely at all evaluated neighborhood sizes, consistent with Theorem 8. To make this more explicit, we have added the following sentence to Section 4.5: "In terms of memory, neighbor-aware aggregation incurs no measurable overhead over PNA (Figure 4)."
>
> ---
> ### (2) Concrete examples of per-neighbor statistics
> The paper touches on this point in three places: the introduction motivates neighbor-aware aggregation with an example from financial domain (identifying the largest per-sender total); Theorem 1 in Section 3.2 formally establishes the expressiveness gap for moment-based aggregators; and Section 4.1 evaluates this capability on a synthetic diagnostic targeting five per-neighbor statistics. However, none of these connections are made explicit in the theory section itself.
>
> To address this, we have extended the paragraph following Theorem 1 with concrete examples, the variance across neighbors of their per-neighbor sums, and the sum across neighbors of their per-neighbor variances, and we noted in the limitations (we added a limitation section, see our next answer) that analogous separations under other aggregator families remain open.
>
> ---
> ### (3) Limitations discussion
> We thank the reviewer for this suggestion. We have added a Limitations and Future Work paragraph after the Conclusion. It discusses two limitations: (i) our theoretical analysis is restricted to the moment-based aggregator family (sum and raw moments of orders 2 through k), within which we show neighbor-aware aggregation is strictly more expressive than single-stage aggregation; extending this result to other aggregation mechanisms remains an open question; and (ii) our empirical evaluation covers financial transaction networks and temporal user-item interaction graphs, and applying MEGA-GNN to broader domains such as property graphs, transportation networks, and cybersecurity datasets is left to future work.
>
> Regarding the reviewer’s suggestion on parallelization, each scatter-reduction step in the neighbor-aware aggregation is fully parallel across segments. As a result, MEGA-GNN has the same parallelism characteristics as standard message-passing GNNs. The only difference is that it performs one additional aggregation per layer. This introduces a small constant-factor overhead in computation, which we already report as approximately 16\% in throughput in Section 4.5. For this reason, we did not include parallelization as a separate limitation.
>
> ---
> ### (4) Edge-type-aware aggregation
>
> The reviewer's suggestion is closely related to a design we present in the paper, MEGA-R-GCNE (Section 4.3.1 and in Appendix A.1, Figure 6). MEGA-R-GCNE  realizes the idea of edge-type-aware multi-edge aggregation. Firstly, edge types are derived from transaction currency on AML benchmark. Then, within each ordered node pair, multi-edges are grouped by their edge type, and the first-stage aggregator is applied separately to each group. Each per-type aggregated representation is then routed through a relation-specific transformation $W_r$ before being passed to the second-stage neighbor aggregation. In this way, edge-type information directly shapes how multi-edges are aggregated, rather than only being available as an input feature.
>
> The empirical results on AML (Figure 3 and Table 2) show that MEGA-R-GCNE consistently outperforms R-GCNE, providing concrete evidence that explicit type-awareness during multi-edge aggregation improves the model.
>
> We note that, in the AML experiments, edge type (transaction currency) is already provided as an input edge feature for all evaluated models alongside timestamp, amount, and payment format (Section 4.3.1), so neighbor-aware aggregation has implicit access to type information.
>
> The reviewer's specific construction, *learnable edge-type embeddings added into edge features*, is a complementary variant of MEGA-R-GCNE. MEGA-R-GCNE injects type information through relation-specific *parameters* ($W_r$ matrices), whereas the suggested construction would inject it through a learned *feature representation* of the edge-type.

---

> > ### Comment · Reviewer_1SzU · 2026-06-28
> >
> > I thank the authors for their thorough and responsive rebuttal. The revised manuscript addresses each of my comments, and I am happy to update my assessment accordingly.
> > On the runtime and memory overhead (point 1), the explicit quantification now  gives the concrete cost/benefit breakdown I was looking for.
> > On limitations (point 3), the clarification on parallelization adequately resolves my concern, and I agree it does not warrant a separate limitation.
> > On edge-type-aware aggregation (point 4), I take the point that my suggested learnable edge-type embeddings are a complementary variant injecting type information through learned features rather than relation-specific parameters; this is a reasonable scoping decision and an interesting direction for future work rather than a required change.
> > I am happy to recommend acceptance of the paper.